# Accelerated cryo-EM structure determination with parallelisation using GPUs in RELION-2

**Dari Kimanius[1†], Björn O Forsberg[1†], Sjors HW Scheres[2\*], Erik Lindahl[1,3\*]**

[1]Department of Biochemistry and Biophysics, Science for Life Laboratory, Stockholm University, Stockholm, Sweden; [2]MRC Laboratory of Molecular Biology, Cambridge, United Kingdom; [3]Swedish e-Science Research Center, KTH Royal Institute of Technology, Stockholm, Sweden

**Abstract** By reaching near-atomic resolution for a wide range of specimens, single-particle cryo-EM structure determination is transforming structural biology. However, the necessary calculations come at large computational costs, which has introduced a bottleneck that is currently limiting throughput and the development of new methods. Here, we present an implementation of the RELION image processing software that uses graphics processors (GPUs) to address the most computationally intensive steps of its cryo-EM structure determination workflow. Both image classification and high-resolution refinement have been accelerated more than an order-of-magnitude, and template-based particle selection has been accelerated well over two orders-of-magnitude on desktop hardware. Memory requirements on GPUs have been reduced to fit widely available hardware, and we show that the use of single precision arithmetic does not adversely affect results. This enables high-resolution cryo-EM structure determination in a matter of days on a single workstation.

**\*For correspondence:** scheres@mrc-lmb.cam.ac.uk (SHWS); erik.lindahl@dbb.su.se (EL)

[†]These authors contributed equally to this work

## Introduction

With the advent of direct-electron detectors and advanced methods of image processing, structural characterisation of macromolecular complexes to near-atomic resolution is now feasible using single-particle electron cryo-microscopy (cryo-EM) (*Cheng, 2015*; *Fernandez-Leiro and Scheres, 2016b*). Although this has caused a rapid gain in its popularity, two technological factors still limit wide applicability of cryo-EM as a standard tool for structural biology.

First, partly due to the steep increase in demand, access to high-end microscopes is limited. This is being addressed with acquisition of new equipment in a large number of departments worldwide, as well as the establishment of shared infrastructures (*Saibil et al., 2015*). Second, processing the large amounts of data produced by these microscopes requires computational hardware that is not directly accessible to many labs. Even at larger centres the computational requirements are so high that cryo-EM now suffers from a computational bottleneck. The work presented here addresses this second problem, to the end of drastically reducing the computational time and investment necessary for cryo-EM structure determination.

A typical cryo-EM data set may constitute hundreds or thousands of images (called micrographs) of a thin layer of vitreous ice in which multiple individual macromolecular complexes (called particles) are imaged. Because radiation damage imposes strict limitations on the electron exposure, micrographs are extremely noisy. Thus, to extract fine structural details, one needs to average over multiple images of identical complexes to cancel noise sufficiently. This is achieved by isolating two-dimensional particle-projections in the micrographs, which can then be recombined into a three-

dimensional structure (*Cheng et al., 2015*). The latter requires the estimation of the relative orientations of all particles, which can be done by a wide range of different image processing programs, such as SPIDER (*Frank et al., 1981*, *1996*), IMAGIC (*van Heel et al., 1996*), BSOFT (*Heymann and Belnap, 2007*), EMAN2 (*Tang et al., 2007*), SPARX (*Hohn et al., 2007*), FREALIGN (*Grigorieff, 2007*), XMIPP (*Scheres et al., 2008*), RELION (*Scheres, 2012a*), or SIMPLE (*Elmlund and Elmlund, 2012*).

These programs also need to tackle the problem that any one data set typically comprises images of multiple different structures; purified protein samples are e.g. rarely free from all contaminants. Multiple conformations, non-stoichiometric complex formation, or sample degradation are all possible sources of additional data heterogeneity. The classification of heterogeneous data into homogeneous subsets has therefore proven critical for high-resolution structure determination and provides a tool for structural analysis of dynamic systems. However, identifying structurally homogeneous subsets in the data by image classification algorithms adds computational complexity, and often increases the computational load dramatically.

An increasingly popular choice for processing cryo-EM data is an empirical Bayesian approach to single-particle analysis (*Scheres, 2012b*) implemented in the computer program RELION (*Scheres, 2012a*). In the underlying regularised likelihood optimisation algorithm, optimal weights for the contribution of different Fourier components to the determination of orientations, as well as to the three-dimensional reconstruction(s), are learnt from the data in an iterative manner, thereby reducing the need for user input. In addition, RELION has proven highly effective in classifying a wide range of structural variation, such as conformational dynamics within protein domains (*Bai et al., 2015*), or of very small sub-populations in large data-sets (*Fernández et al., 2013*). Unfortunately, the regularised likelihood optimisation algorithm that underlies these calculations is computationally demanding. We estimate that a recent 3.7 Å structure of a yeast spliceosomal complex (*Nguyen et al., 2016*) required more than half a million CPU hours of classification and high-resolution refinement. Computations of this magnitude require the use of high-performance computing clusters with dedicated staff, and restrict the exploration of new image processing schemes.

The introduction of hardware accelerators, such as graphics processors (GPUs), has recently transformed other scientific fields where computation was a bottleneck. To exploit this type of hardware, substantial redesigns of algorithms are required to make many independent tasks simultaneously available for computation, which is known as exposing (low-level) parallelism. However, the possible gain is equally substantial; together with commodity hardware it has been a revolution e.g. for molecular dynamics simulations (*Salomon-Ferrer et al., 2013*; *Abraham et al., 2015*), quantum chemistry (*Ufimtsev and Martínez, 2008*), and machine learning (*Jia et al., 2014*).

GPU acceleration of computationally expensive algorithms has also been performed in other cryo-EM software, with varying success and subsequent impact in the field. Many attempts have achieved significant improvement in performance through acceleration of only a few subroutines (*Li et al., 2010*; *Tagare et al., 2010*; *Li et al., 2013*; *Hoang et al., 2013*), relieving major bottlenecks. The overall software architecture and data structures were however not modified to better suit the hardware. In contrast, other tools were designed or fundamentally reformulated with a particular hardware in mind (*Li et al., 2015*; *Zhang, 2016*); this tends to lead to more substantial long-term performance benefits, and for some steps the accelerated implementations have fully replaced CPU codes. Historically, RELION (like most alternatives) has scaled to the large resources it needs by using higher-level parallelism, where computational tasks are divided into subsets of particle images in each iterative refinement step. However, lower-level core computations on individual particle images in RELION have remained serialised since its introduction more than four years ago.

Here, we describe a new implementation of the regularised likelihood optimisation algorithm in RELION that uses GPUs to address its computational bottlenecks. We have chosen to implement our increased parallelism in CUDA, a programming language provided by NVIDIA. The CUDA language currently dominates the GPU computing market, and provides a stable programming environment with a rich C++ interface. We also utilise a number of libraries provided within the CUDA framework, such as cuFFT for fast Fourier transforms (FFTs), and CUB/thrust for sorting and other standard functions. In addition to high-end professional cards there is wide availability of cheap consumer hardware that supports CUDA, which provides outstanding value for many research groups. However, the acceleration and parallelisation approaches are general and should be possible to port to other architectures in the future.

The present acceleration of relion addresses the most computationally intensive steps in a typical image processing workflow. This includes classification of data into structurally homogeneous subsets (2D or 3D classification) and high-resolution refinement of each homogeneous such set of particles (3D auto-refine). In addition, we describe an improved algorithm for semi-automated selection of particles from micrographs (*Scheres, 2015*), this too targeting GPUs. Memory requirements have been reduced to fit widely available consumer graphics cards, and we show that the current adaptation to use single precision floating-point arithmetic does not cause loss of resolvable detail in the final structures. These developments enable high-resolution cryo-EM structure determination in a matter of days on individual workstations rather than relying on large clusters.

## Methods

### Regularised likelihood optimisation

The regularised likelihood optimisation in relion uses an Expectation-Maximization algorithm (*Dempster et al., 1977*) to find the most likely 3D density map for a large set of 2D particles images with unknown orientations, under the prior expectation that the 3D map has limited power in the Fourier domain. This iterative algorithm involves two fundamental steps at every iteration. In the expectation, or E-step, one calculates probability distributions for the relative orientations of all particles based on the current estimate of the 3D map. In the subsequent maximization, or M-step, one updates the estimate for the 3D map. Intuitively, these two steps represent alignment of the particles with respect to a common 3D reference, and reconstruction of a new reference map, respectively. The algorithm is typically started from an initial 3D reference map at low resolution, and is guaranteed to increase the likelihood of the 3D map given the data and the prior at every iteration, until converging onto the nearest local minimum. In the presence of structural variability in the data set, multiple 3D reference maps, or 3D classes, can be refined simultaneously. In this case, the class assignments of the particles are not known either. RELION also allows the refinement of multiple 2D references, or 2D classes. In that case, only in-plane orientations are sampled.

Within a single iteration, the E-step requires four major computational stages (*Figure 1*): (i) The current 3D reference map is projected along many orientations; (ii) the difference between these reference projections and every particle image is calculated for each orientation and for each sampled translation; (iii) all these differences are converted to probability-weights; and (iv) those weights are used to update a running sum of the back-projected particle images in 3D Fourier space. In practice, to limit the number of operations, stages (i) and (ii) are performed twice for every particle. A first

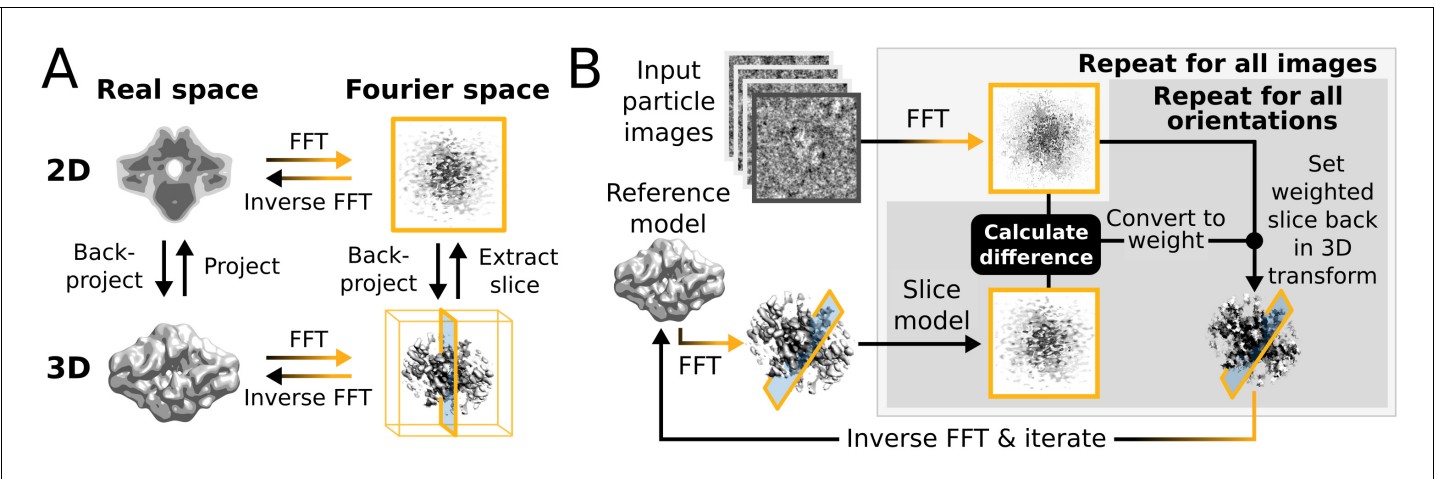

**Figure 1.** High level flowchart of RELION. (**A**) Operations and the real vs. Fourier spaces used during (**B**) image reconstruction in RELION. Micrograph input and model setup use the CPU, while most subsequent processing steps have been adapted for accelerator hardware. The highlighted orientation-dependent difference calculation is by far the most demanding task, and fully accelerated. Taking 2D slices out of (and setting them back into) the reference transforms has also been accelerated at high gain. The inverse FFT operation has not yet been accelerated, but uses the CPU.

pass examines all orientations using a relatively coarse orientational sampling, and a second pass re-examines the regions of orientational space with significant probabilities at an increased sampling density. Still, millions of orientations and translations are typically compared for each particle at every iteration. Converting the resulting differences to weights is a relatively cheap operation as each image-orientation pair at this stage is represented by a single scalar value. The back-projection in operation (iv) is more demanding since it is again necessary to work with all pixels of each image in multiple orientations. After the E-step has been evaluated for all (typically tens to hundreds of thousand) particles, in the M-step the running sum in 3D Fourier space is transformed into an updated 3D reference map for the next iteration. The corresponding reconstruction algorithm is expensive in terms of computer memory, but it typically needs orders of magnitude less time than the E-step because it is only performed once for every iteration. Therefore, operations (i), (ii) and (iv) of the E-step have dominated execution time in previous versions of RELION.

## Extracting parallelism for accelerators

Modern accelerator processors - such as GPUs - achieve very high floating-point performance by incorporating a large set (thousands) of very simple and streamlined functional units instead of the handful advanced general-purpose cores in a normal (CPU) processor. In previous versions, RELION has scaled over multiple CPU cores by using separate processes for independent particle images. GPUs however require much broader low-level parallelism to increase performance substantially. Our implementation of RELION therefore required a reformulation to expose additional available parallelism. Since most of the computation time is spent on the E-step for a typical application, we focussed our acceleration efforts on this part. By treating multiple reference maps, all relative image translations and orientations, and even individual image pixels as parallel tasks (*Figure 2*), we created sufficient low-level parallelism to provide outstanding performance on accelerator hardware such as GPUs.

Formulating and enabling the available parallelism in RELION for GPUs did present specific challenges. In operations (i) and (ii) of the E-step, it would for instance require very large amounts of fast memory to pre-calculate projections of the 3D reference maps for all examined orientations. Pre-calculation could potentially increase re-use of data on the GPU, but it would impose severe limitations on the number of simultaneously examined references, input image sizes, and sampling granularity.

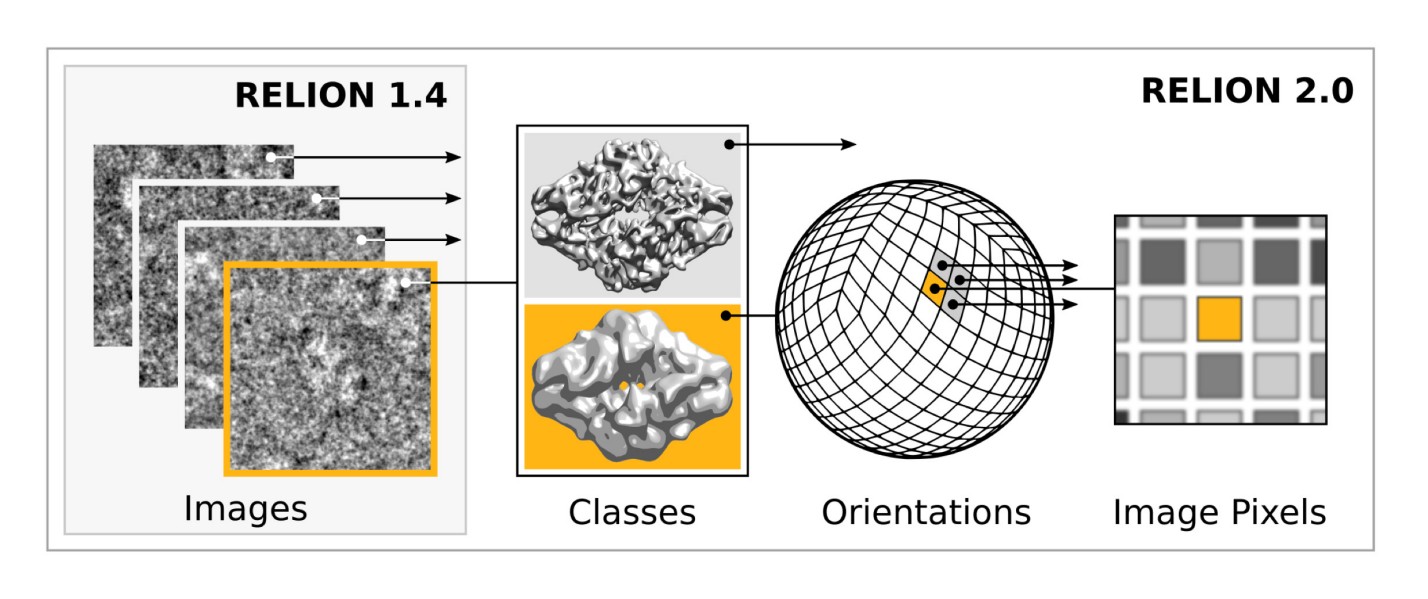

**Figure 2.** Extensive task-level parallelism for accelerators. While previously relion only exploited parallelism over images (left), in the new implementation classes and all orientations of each class are expressed as tasks that can be scheduled independently on the accelerator hardware (e.g. GPUs). Even individual pixels for each orientation can be calculated in parallel, which makes the algorithm highly suited for GPUs.

Instead, like the CPU code, our implementation stores a two-fold oversampled Fourier transform of each reference in GPU memory, and 2D slices (along any orientation) are extracted only when needed. This alleviates limitations imposed by hardware, and makes it possible to improve performance through the use of so-called GPU *textures*. Textures store data that require fast lookup, and the corresponding *texture units* provide support for performing non-integer pixel interpolation of texture data in a single instruction with only marginal loss of precision. In graphics applications, textures are used to efficiently rescale image data. In RELION, they turn out to be well-suited for the resampling operations when taking slices from the 3D Fourier transform. The on-the-fly extraction of 2D Fourier slices is combined with the very broad parallelism of calculating squared differences between all pixels of all particle images and the reference projections in all orientations and all translations. For the technically interested reader, details of how we exploit this parallelism in our implementation of the squared difference calculation are presented in Appendix I. Technical details of how we also accelerated operation (iv) of the E-step, the back-projection of 2D images into the oversampled 3D Fourier transform, are presented in Appendix II. When combined, the E-step can now be evaluated in a fraction of the runtime previously needed.

Finally, like most scientific software, the CPU version of relion has historically used double precision, and merely recompiling in single does not improve performance. However, all GPUs provide higher performance for single precision, and the dedicated texture units can typically only perform reduced-precision operations. For these reasons we also reformulated our implementation of the E-step to be less precision-sensitive, making it possible to execute these calculations in single precision without adverse effects on the final results. As we did not accelerate the M-step, this part of the algorithm remains on the CPU and is still executed in double precision.

## Semi-automated picking

RELION also implements a template-based particle selection procedure, which calculates a probability measure (the R-value) for each pixel in the micrograph to signify the likelihood that it is the location of any of the provided templates (*Scheres, 2015*). The R-value map of a micrograph considers all possible rotations of each template, and is subsequently used in a peak-search algorithm that locates particles within the original micrograph (*Figure 3*). These calculations are performed in Fourier space, where they are highly efficient (*Roseman, 2003*). In fact, they are so fast that their execution time becomes negligible compared to the time spent performing FFTs to transform image objects between real and Fourier spaces. Consequently, even though reference templates are also treated as independent tasks to increase parallelism in the GPU version, a much larger gain is found at the level of template rotations, through parallel execution of FFTs. For example, when using 5-degree incremental template rotations, 72 such inverse FFTs are now performed concurrently on the GPU through the cufft cuda library. The size of these FFTs is now also padded automatically, since substantial performance penalties can occur if the transform size includes any large prime factors.

## Results

## Acceleration of regularised likelihood optimisation
### Performance
The performance of our implementation on a workstation equipped with modern GPUs can exceed that of hundreds of CPU cores (*Figure 4*). This is most prominent for increasing numbers of pixels, orientations and classes, due to the increased low-level parallelism RELION-2 has been designed to use efficiently. Therefore, calculations where many classes and orientations need to be sampled, e.g. 3D classifications over multiple classes and with fine sampling of orientations, experience the greatest gain from the acceleration (*Figure 4*). Traditionally such large problems have required cluster-size resources with high-performance interconnects for fast communication. As seen in the performance benchmarks, RELION-2 makes it possible to run many of these calculations even on workstations or low-cost desktop solutions (*Figure 4*). GPU hardware evolves rapidly, but appendix XI contains a few recommendations about the solutions we currently think are most cost-efficient. For sufficiently large computational problems, relion's processing time scales linearly with increased number of classes, but since the extra calculations are much faster with the GPU-enabled version the relative advantage is larger the more classes are used (*Figure 4D*). In practice we believe this will

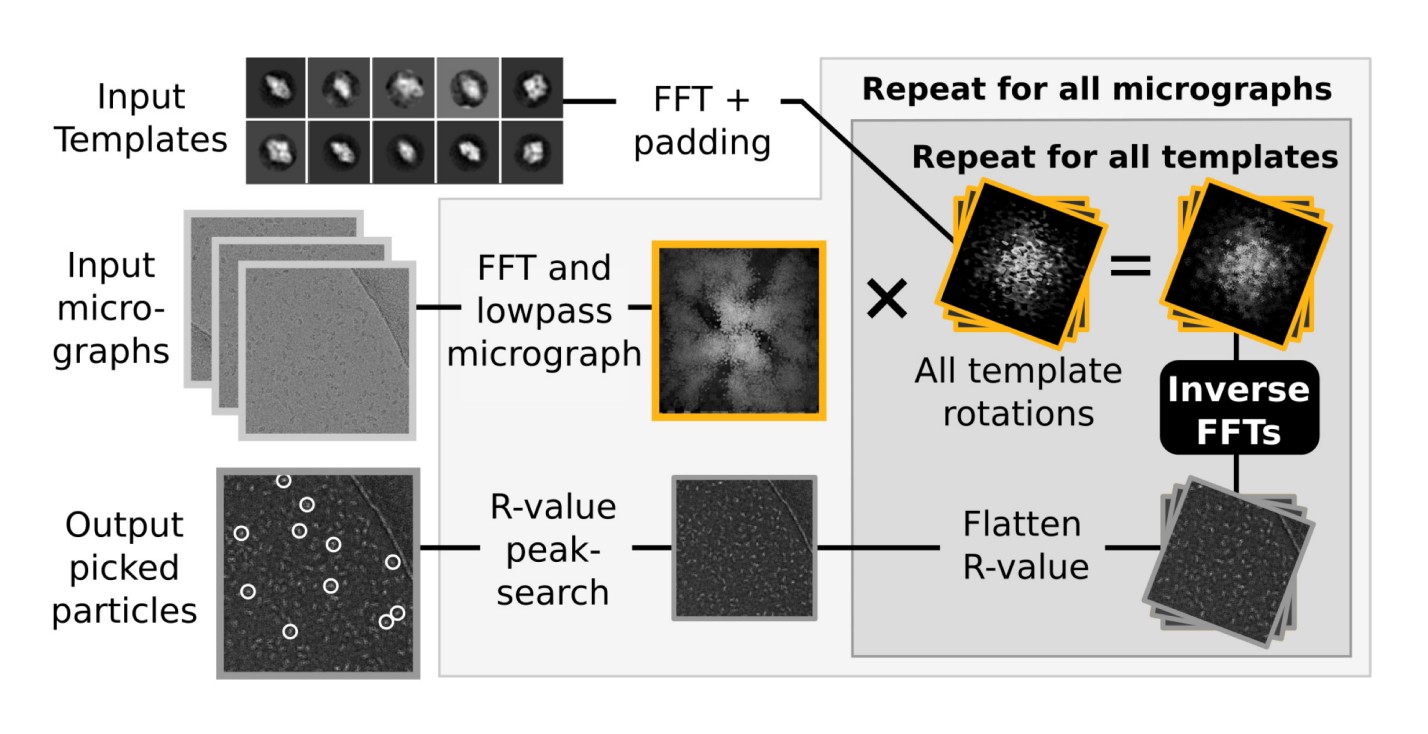

**Figure 3.** Semi-automated particle picking in RELION-2. The low-pass filter applied to micrographs is a novel feature in RELION, aimed at reducing the size and execution time of the highlighted inverse FFTs, which accounts for most of the computational work. In addition to the inverse FFTs, all template- and rotation-dependent parallel steps have also been accelerated on GPUs.

make it more common to use very large numbers of classes in 3D classification. Further scrutiny reveals that the calculations still performed on the CPU actually dominate both the execution and scaling even for the GPU version (*Figure 4E*), which indicates new bottlenecks are now limiting scaling - we intend to focus on these parts of the code for future improvements.

## Limited precision and accuracy

RELION has used double precision arithmetic since its first release in order to be as accurate as possible, at the cost of increased memory requirements. While there are professional GPUs with good double precision performance, the consumer market is dominated by visualisation and gaming applications, and for this reason cheap hardware only provides good performance for single precision. Even for professional hardware, the performance is better with single precision, although the difference is smaller. This makes it highly desirable to use single precision arithmetics wherever possible. In addition to much better floating-point throughput, single precision calculations reduce the memory requirements by a factor two, and make it possible to employ textures for image rotation. The relative advantage of single precision can thus be much higher on GPUs compared to CPUs. Because the required precision depends on the algorithms used in the application, part of the development of RELION-2 involved the evaluation of reconstructed quality when using single precision. We evaluate this primarily by examining the agreement of refinements results, characterising both the reconstructed volume and the image orientation assignment statistics. As image orientations are analogous to a location and rotation on the unit sphere, we compared refinement results of the EMPIAR 10028 dataset by the distribution of angular differences for all images. Two double-precision CPU refinements (using different random seeds) produce distributions where 81% of images fall within $1°$ of the other. Given that this distribution is modulated by the sine of the angle, this is a very close agreement. The single-precision GPU implementation finds 82% of images within the same $1°$ tolerance. (*Figure 5—figure supplement 1*). In other words, the differences due to random seeds are at least as large as any systematic variation between the CPU and GPU versions. While a small

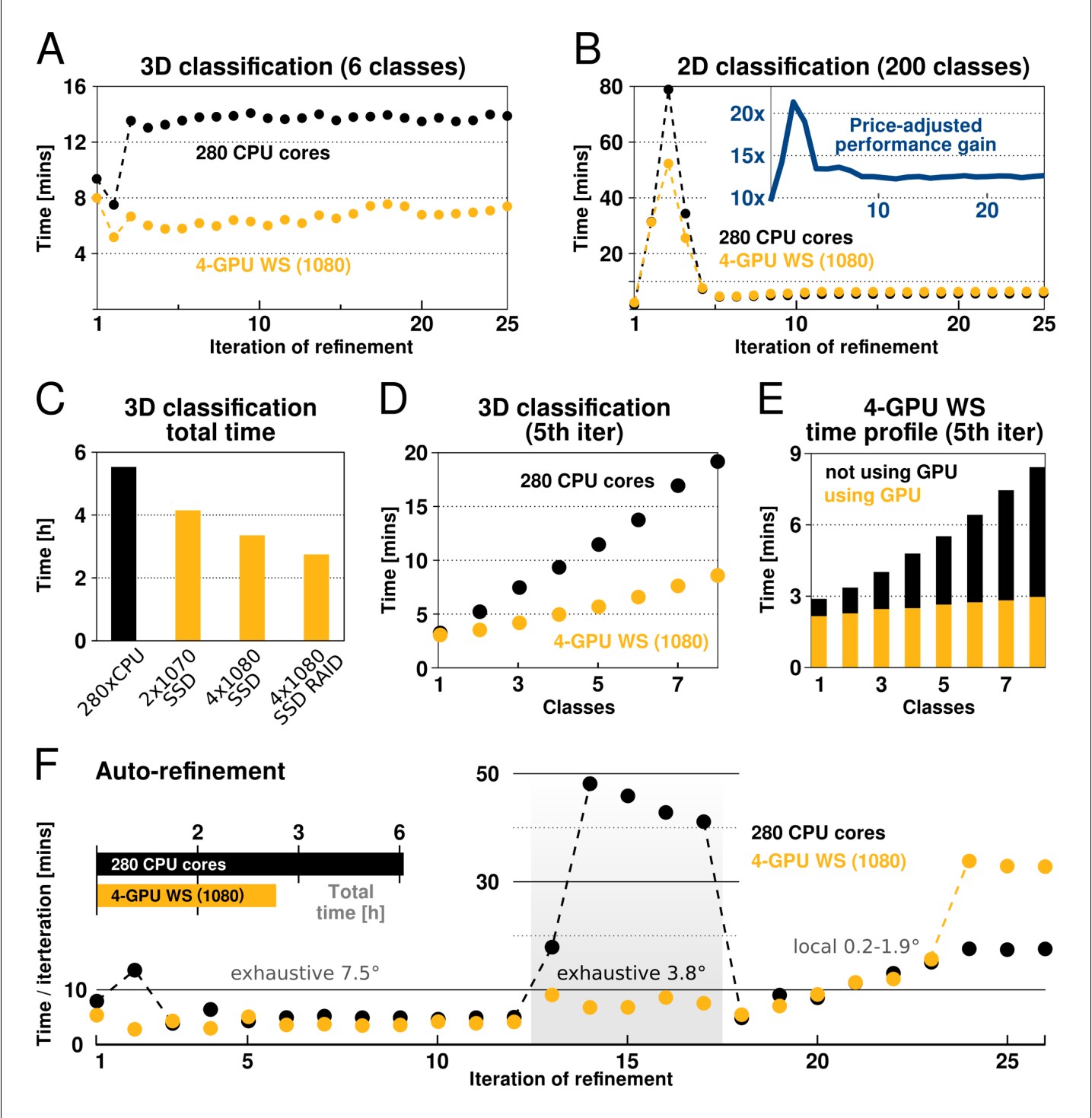

**Figure 4.** RELION-2 enables desktop classification and refinement using GPUs. EMPIAR (*Iudin et al., 2016*) entry 10028 was used to assess performance, using refinements of 105 k ribosomal particles in $360^2$-pixel images. (**A**) A quad-GPU workstation easily outperforms even a large cluster job in 3D classification. (**B**) In 2D classification, the GPU desktop performs slightly better in the first few iteration and then provides performance equivalent to the 280 CPU cores. (**C**) Total time for 25 iterations of 3D classification for a few different hardware configurations. (**D**) Additional classes are processed at reduced cost compared to CPU-only execution, due to faster execution and increased capacity for latency hiding. (**E**) With increasing number of classes, the time spent in non-accelerated vs accelerated execution increases. (**F**) The workstation also beats the cluster for single-class refinement to high resolution, despite the generally lower degree of parallelism. This is particularly striking for the finer exhaustive sampling at 3.8° due to the GPU's ability to parallelise the drastically increased number of tasks.

loss of precision was observed e.g. in the fast interpolation (texture) intrinsics, the assignments of image orientations are similar enough to not influence the final reconstruction quality (*Figure 5*). In contrast, single-precision execution of the iterative gridding algorithm that underlies the reconstruction in the M-step (*Scheres, 2012a*) did exhibit a notable deviation from the double-precision version when first tested. Therefore, we opted for a hybrid implementation of the algorithm: The computationally demanding slice projections, probability calculations, and back-projections in the E-step are performed in single precision on the GPU, while the reconstruction in the M-step remains in double precision on the CPU. Because the M-step is partially responsible for the new bottlenecks apparent in *Figure 4E* this is a candidate for additional future optimisation, but for now the hybrid approach provides a good compromise.

## Disk and memory considerations

relion has traditionally required large amounts of memory. Fortunately its peak use is however *not* during the accelerated, computationally most intensive, parts of the algorithm. Rather, memory use peaks during the M-step, which is executed on the CPU using the gridding reconstruction algorithm mentioned above. The amount of available GPU memory still remains a limitation, as it determines the capacity for storage of the oversampled Fourier transforms of one or more references. This is of particular concern for larger and higher-resolution structures, which require more memory to be faithfully represented. When resolving detail at the Nyquist frequency, due to a twofold oversampling (associated with gridding during reconstruction), relion requires memory corresponding to twice the cube of the image dimension. For example, when using $400^2$-pixel particle images, the required grid size is $800^3$, which becomes $\sim 2$ GB per class, since each value requires 4 bytes in single precision. Moreover, as the reconstructed object also needs to be accommodated, this number

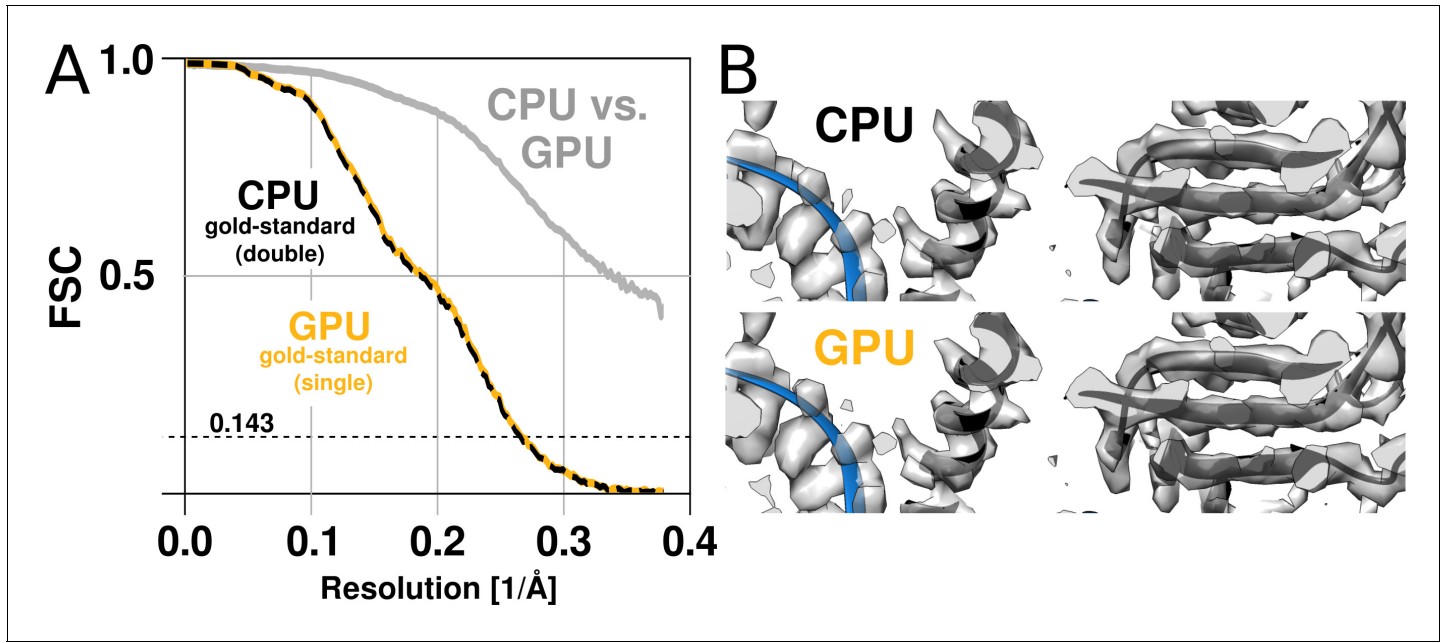

**Figure 5.** The GPU reconstruction is qualitatively identical to the CPU version. (**A**) A high-resolution refinement of the Plasmodium falciparum 80S ribosome using single precision GPU arithmetic achieves a gold-standard Fourier shell correlation (FSC) indistinguishable from double precision CPU-only refinement (previously deposited as EMD-2660). The FSC of full reconstructions comparing the two methods shows their agreement far exceeds the recoverable signal (grey), and as shown in *Figure 5—figure supplement 1* the variation in angle assignments match the differences between CPU runs with different random seeds. (**B**) Partial snapshots of the final reconstruction following post-processing, superimposed on PDB ID 3J79 (*Wong et al., 2014*).

The following figure supplement is available for figure 5:

**Figure supplement 1.** The CPU and GPU implementations provide qualitatively identical distributions of image orientations.

is effectively multiplied by 2.5. In practice, memory use with image sizes up to $400^2$ indicate that at most 6 GB of GPU memory is needed to perform refinement to Nyquist (*Figure 6*). Classification using 3D references is usually performed at resolutions much lower than the Nyquist limit, and for this reason its memory requirements are typically much lower.

To enable efficient evaluation and good scaling on GPUs, several new methods to manually manage data efficiently have been implemented. Lower levels of parallelism are coalesced into larger objects using customised tools, which results in more efficient use of memory (see appendices I-II for details). In addition, because of the much improved performance, multiple tasks have become limited by how fast input data can be read from disk. Therefore, we now find it highly beneficial to explicitly cache data on local solid state devices (SSDs), as has also been observed for GPU-accelerated CTF estimation (*Zhang, 2016*). To allow this in a straightforward way, RELION-2 features the ability to automatically copy data sets to fast local disks prior to refinement, which further increases performance during less computationally intensive steps, such as 2D classification.

## Acceleration of automatic particle picking
### Low-pass filtering of micrographs
Even after parallelisation and acceleration on GPUs, the cross-correlation-based particle selection in relion is dominated by computing many large inverse FFTs (*Figure 3*), as has been observed previously for similar methods (*Castaño-Díez et al., 2008*). Reducing their size is thus the most straightforward way to further improve execution performance. Reference templates are typically subject to low-pass filtering, and for this reason we investigated the possibility to apply a similar filtering to all micrographs, which reduces computations by discarding high-frequency information.

We found little difference in the particles selected when discarding resolution information in micrographs beyond that of search templates. While intuitively straightforward, this conclusion drastically reduces the size of FFT grids and subsequent computations, which provides large acceleration at virtually no quality loss. The low-pass filtering also significantly reduces the amount of memory required for particle selection, which permits parallelism to target hardware like desktop workstations more efficiently.

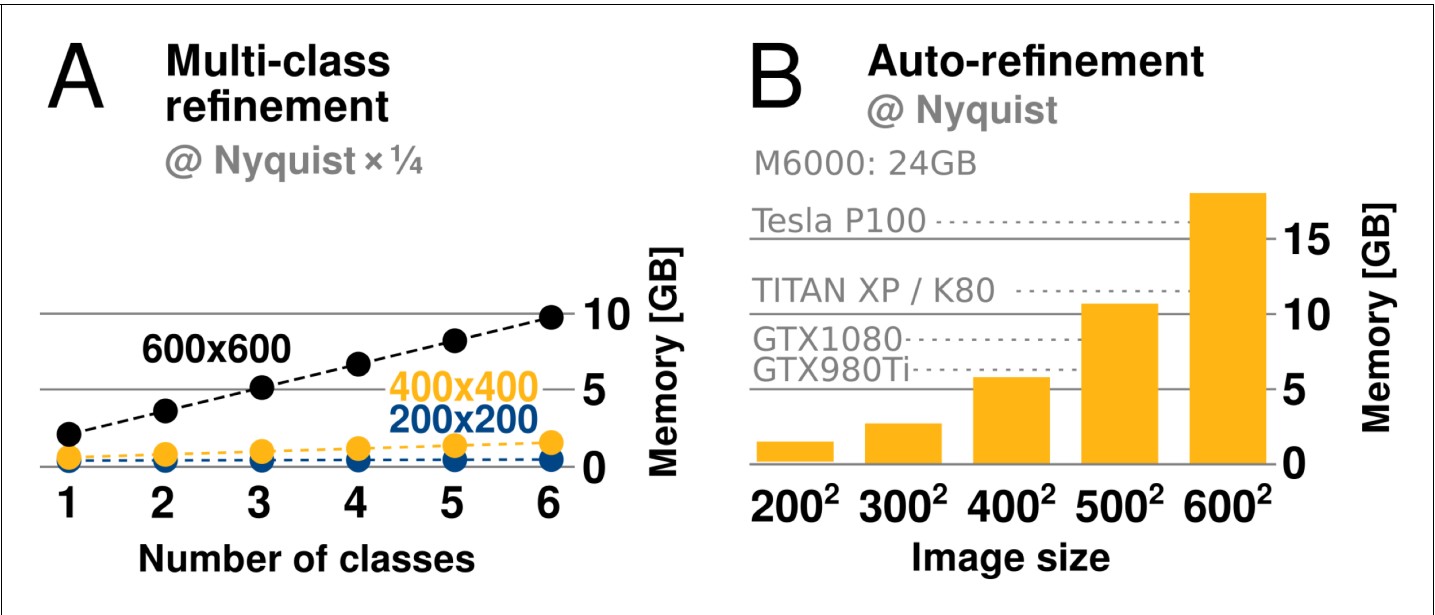

**Figure 6.** GPU memory requirements. (**A**) The required GPU memory scales linearly with the number of classes. (**B**) The maximum required GPU memory occurs for single-class refinement to the Nyquist frequency, which increases rapidly with the image size. Horizontal grey lines indicate avaliable GPU memory on different cards.

## Autopicking performance

We tested both picking speed and quality of picked particles of RELION-2. In an initial test, a single $4096^2$-pixel micrograph containing ribosomes at 1.62 Å/pixel was processed against 8 templates with 5 degree angular sampling and no low-pass filtering. This took 675 s to evaluate on a CPU-only workstation (i7–5960X, using 1 thread merely to provide a per-core performance). When applying low-pass filtering to 20 Å, this time is reduced to 39 s, i.e. by a factor ∼17. When using a single consumer-level GPU (GTX 1080) with a single CPU thread, execution is further reduced to just 0.73 s, or an additional factor ∼54. Each GPU added to the workstation can therefore now process ∼940 micrographs in the same time previously required to process just 8 micrographs (1 per available core) as shown in *Figure 7*.

We further evaluated the quality of filtered selection according to the *β*-galactosidase benchmark (EMPIAR entry 10017) used in the original implementation in RELION-1.3 (*Scheres, 2015*). This data set consists of 84 micrographs of $4096^2$ pixels (1.77 Å/pixel), and comes with coordinates for 40,863 particles that were manually selected by Richard Henderson. The latter were used for comparison with our autopicking results, with a center cutoff distance of 35 pixels for particles to be considered identical (*Table 1*). Filtered selection did not decrease the quality of the results, but provided a just-so-slightly increased recall without increasing the false discovery rate (FDR, see e.g. (*Langlois and Frank, 2011*) for definitions of recall and FDR). When filtering and GPU-acceleration are combined, a single GPU provides roughly 120 times the performance of an 8-core desktop, and the desktop can easily be equipped with quad GPUs for about 500x performance gain. In fact, similarly to the regularised likelihood optimisation, the semi-automated particle picking can become limited by disk access unless data is read from an SSD.

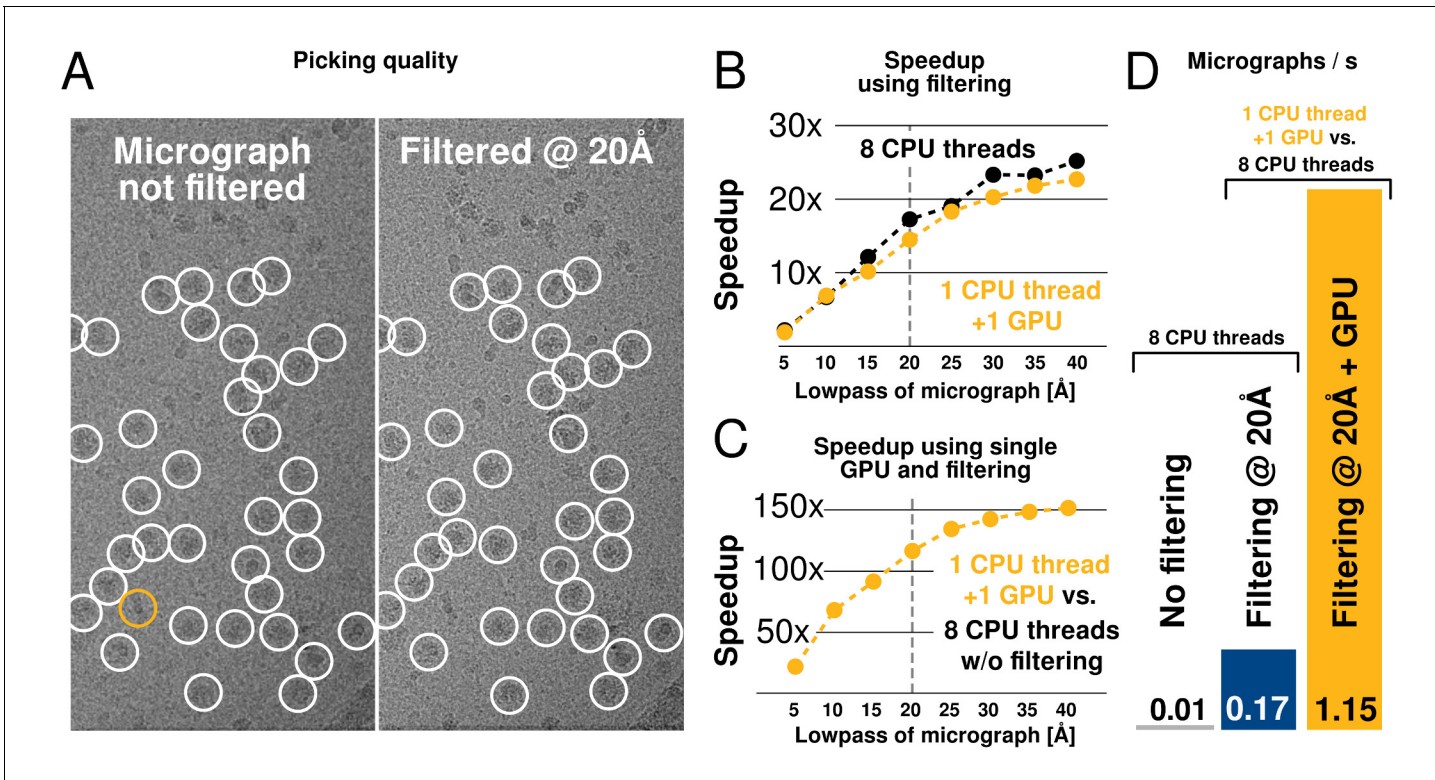

**Figure 7.** Low-pass filtering and acceleration of particle picking. (**A**) Ribosomal particles were auto-picked from representative $4096^2$-pixel micrographs collected at 1.62 Å/pixel using four template classes, showing near-identical picking with and without low-pass filtering to 20 Å. The only differing particle is indicated in orange, and likely does not depict a ribosomal particle. (**B–C**) Despite near-identical particle selection, performance is dramatically improved. (**D**) Filtering alone provides almost 20-fold performance improvement on any hardware compared to previos versions of relion, and when combined with GPU-accelerated particle picking the resulting performance gain is more than two orders of magnitude using only a single GPU (GTX 1080).

**Table 1.** Quality and speed of autopicking for the β-galactosidase benchmark. Comparing the CPU version with the GPU version using increasing levels of low-pass filtering yields progressively higher recalls at similar FDRs. The GPU version yields identical results to that of the CPU version, but at a much reduced computational costs. Filtering does not depend on GPU-acceleration, and will perform similarly using only CPUs.

| Code | Filter | # picked | Recall | FDR | Time | Performance in |
|------|--------|----------|--------|-----|------|----------------|
| | (Å) | particles | | | (s/micrograph) | CPU core units |
| **CPU** | **none** | **54,301** | **0.88** | **0.34** | **1,227** | **1** |
| GPU | none | 54,325 | 0.88 | 0.34 | 10 | 122 |
| GPU | 5 | 55,629 | 0.90 | 0.34 | 5.8 | 211 |
| GPU | 10 | 55,886 | 0.90 | 0.34 | 2.1 | 584 |
| GPU | 15 | 56,450 | 0.92 | 0.33 | 1.6 | 766 |
| GPU | 20 | 57,361 | 0.95 | 0.33 | 1.3 | 943 |

## A complete workflow for β-galactosidase

To illustrate the impact of our GPU implementation and show how it can alter practical work, we chose to re-analyse the EMPIAR-10061 dataset of β-galactosidase (*Bartesaghi et al., 2015*) using RELION-2. This represents the largest presently available dataset in the empiar database, and provides a realistic challenge. We performed an entire processing workflow, including initial beam-induced motion correction in UNBLUR (*Grant and Grigorieff, 2015*), CTF estimation in Gctf (*Zhang, 2016*), and finally RELION-2 was employed for automated particle picking, 2D and 3D classification, movie-refinement, particle polishing (*Scheres, 2014*) and high-resolution auto-refinement on a single workstation with four GTX 1080 cards. Calculating a map to 2.2 Å resolution (*Figure 8A*) took a few days – comparable to, if not faster than, the time required acquire this amount of high-quality data. *Figure 8B* shows an overview of the most computationally demanding steps during this processing. The parts of the workflow that have been GPU-accelerated no longer dominate execution, but this exposes other new bottlenecks. In particular, steps that involve reading large movie

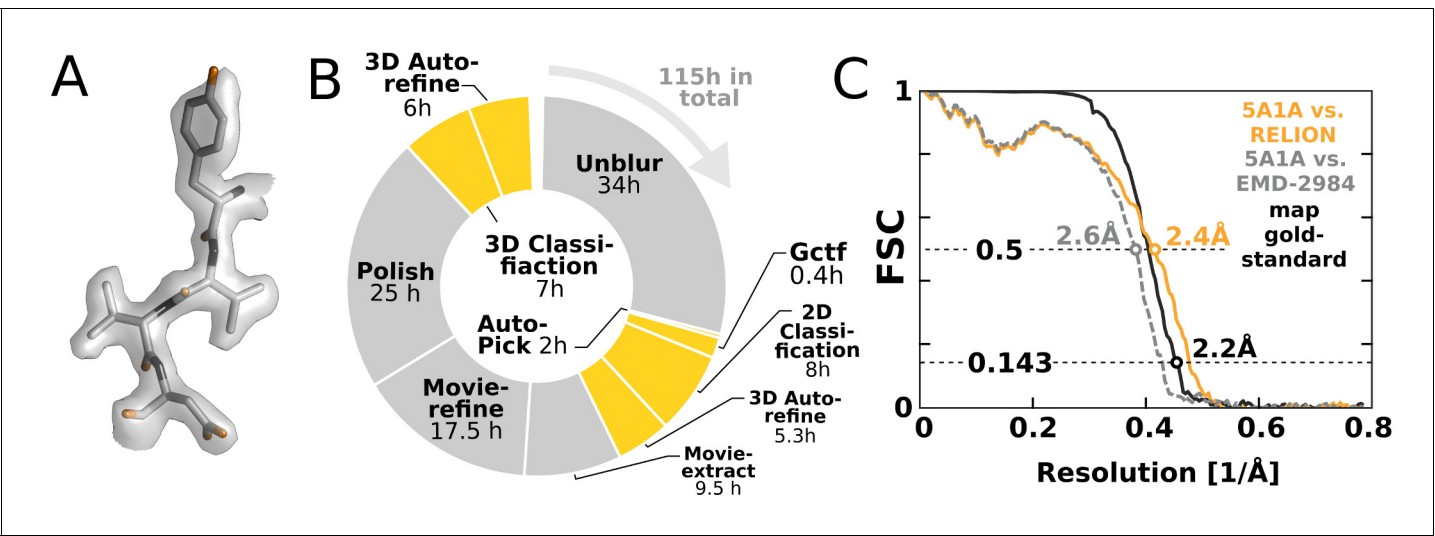

**Figure 8.** High-resolution structure determination on a single desktop. (**A**) The resulting 2.2 Å map (deposited as EMD-4116) shows excellent high-resolution density throughout the complex. (**B**) The most time-consuming steps in the image processing workflow. GPU-accelerated steps are indicated in orange. The total time of image processing was less than that of downloading the data. (**C**) The resolution estimate is based on the gold-standard FSC after correcting for the convolution effects of a soft solvent mask (black). The FSC between the relion map and the atomic model in PDB ID 5A1A is shown in orange. The FSC between EMD-2984 and the same atomic model is shown for comparison (dashed gray).

files from disk become problematic. We also note that due to the rigid nature of the $\beta$-galactosidase complex used for this benchmark, only a single 3D classification was performed. This is not representative for many other use cases: typically 3D classification is repeated multiple times to identify, sort, and isolate structural heterogeneity. In such a scenario, the impact of the GPU acceleration is even larger, as increased or multiple 3D classifications would still not dominate the complete workflow.

## Discussion

We present a GPU-enabled implementation of RELION-2, as a first step to address current and future needs for large and expedient computations in the field of cryo-EM structure determination. The principal benefits drawn from the presented work are twofold. First, the nature of progress in scientific applications is to continually re-evaluate and examine data in many different ways. With ease of re-processing data, the threshold for trial, error and successive improvement of existing methods is now markedly lowered. Second, the order-of-magnitude speedups make it possible to get by with much less hardware for cryo-EM processing, in most cases even desktops. This removes a computational bottleneck for large labs, and enables any group to perform their own reconstruction without access to supercomputers.

In the next few years, larger data sets and image sizes are expected, as well as new methods that require expedient processing of large data sets. The large reduction in computational costs opens up the possibility to perform more ambitious computational analyses without increasing the investments. For example, the favourable scaling of performance we observed for multi-class refinements will make it feasible to use many more classes than was practical before, which will lead to better descriptions of conformational diversity in flexible molecules. Additionally, with even faster algorithms and hardware it might soon be possible to perform highly automated, on-the-fly, structure determination while data acquisition is ongoing. In anticipation of these developments, RELION-2 already implements a pipelined approach for automated execution of pre-determined image processing workflows (*Fernandez-Leiro and Scheres, 2016a*).

While the new GPU implementation has removed many of the previous computational bottlenecks in relion, the large speedup has exposed several new areas of the code that can now dominate execution time, such as data input/output and the reconstruction step during iterative image refinement. Although these parts of the algorithm were previously insignificant, in some cases they now collectively account for roughly 50% of total execution time. These parts of the code will see benefit from further modifications. Future work will e.g. strive to further generalise parallelism such that performance is less dependent on the type of refinement performed, as sufficient parallelism is always available within the relion core algorithm. Memory requirements on the GPU are also expected to be reduced further, so that larger image sizes and more classes can be handled to higher resolution.

With the current implementation, cryo-EM structures to near-atomic resolution can be calculated in a matter of days on a single workstation, or hours on a GPU-cluster. Nevertheless, the aim of the current adaptations is not to present a final solution to computational needs in RELION; while the present version achieves excellent speedup on a wide range of low-cost systems, we expect the acceleration to improve both in performance and coverage. Generalising the low-level parallelism described here to vectorised CPU calculations, and possibly an open GPU language like OpenCL, will constitute little more than translating this parallelism to new instructions. This is something we intend to pursue in the future. As such, RELION-2 represents a new incarnation of an existing algorithm, which is intended to be developed far further in the following years. Meanwhile, we hope that the current implementation will have as much impact in the broader community as it is already having in our labs.

## Materials and methods

### Availability

RELION 2 is both open source and free software, distributed under the GPLv2 licence. It is publicly available for download through http://www2.mrc-lmb.cam.ac.uk/relion.

## Data sets and hardware specifications

The ribosome data used for the 3D classification and refinement in *Figures 4* and *5* correspond to EMPAIR entry 10028 (*Wong et al., 2014*). For autopicking, empiar entry 10017 was used (*Scheres, 2015*). The complete workflow for $\beta$-galactosidase used EMPIAR entry 10061 (*Bartesaghi et al., 2015*), and the reconstructed map was deposited in the EMDB (EMD-4116). In all cases where different hardware and/or software implementations were compared, identical refinement parameters were used, including the random seed provided to RELION. In cases where identical hardware and software were repeatedly used to examine variability, different seeds were used. The nondeterministic scheduling and summations of GPU-enabled execution can introduce some minor noise in results, but all runs achieve convergence and variations in final results are well within estimated errors and resolution of the raw data.

The acceleration in RELION-2 works with any NVIDIA GPU of compute-capability 3.5 or higher, which covers all models launched the last three years. Version 7.0 of the CUDA toolkit is also required, and was used for all presented results. CPU performance was benchmarked with a cluster of 10 compute nodes equipped with dual Xeon E5-2690v4 CPUs ($2\times14$ physical cores, for a total of 280 cores) running at 2.6 GHz with 128 GB memory. While we refer to the physical core count when describing hardware, hyperthreading was enabled and used for all benchmarks (i.e., starting 560 threads on the 10 nodes) since it improves performance on the CPU side slightly. Two different workstations were used for GPU benchmarks. First a cost-efficient desktop with a single Core i7–6700 K (four cores, 4 GHz), dual GTX 1070 GPUs, and a 500 GB SSD disk. Second, a workstations equipped with a single Core i7–5960 X CPU (8 cores, 3GHz), four GTX 1080 GPUs, and either a single SSD or two configured in RAID0 for higher bandwidth. Both workstations had 64 GB of memory, and two CPU threads were used for each GPU to improve utilisation. In all cases hyper-threading was utilised to the fullest extent possible to improve CPU performance.

## $\beta$-galactosidase image processing

Super-resolution $8k \times 8k$ micrograph movies with 38 frames were submitted to initial beam-induced motion correction using UNBLUR (*Grant and Grigorieff, 2015*). The resulting average micrographs were used for CTF estimation in Gctf (*Zhang, 2016*). Autopicking with six templates yielded an initial data set of 130,375 particles, which were subjected to reference-free 2D classification using 200 classes. This initial classification was done using $4\times$ downscaled particles (with a pixel size of 1.274 Å and a box size of 192 pixels). Selection of the 75 best classes resulted in 120,514 particles. All subsequent calculations were performed using $2\times$ downscaling (resulting in a pixel size of 0.637 Å and a box size of 384 pixels). The selected particles were subjected to an initial 3D auto-refinement that used PDB ID 3I3E (*Dugdale et al., 2010*) as an initial model. Subsequent movie-refinement (with a running average of 7 movie frames and a standard deviation of 2 pixels on the translations) was followed by particle polishing (using a standard deviation of 1000 pixels on the inter-particle distance). The resulting shiny particles were submitted to a single round of 3D classification with exhaustive 7.5-degree angular searches and eight classes. Selection of the seven best classes yielded a final data set of 109,963 particles, which were submitted to 3D auto-refinement. The final resolution was estimated using phase-randomisation to account for the convolution effects of a solvent mask on the FSC between the two independently refined half-maps (*Chen et al., 2013*). This mask was generated by binarisation of a 15 Å low-pass filtered version of the reconstructed map, with addition of a five-pixel wide cosine-shaped soft edge. FSC curves between the model and the solvent-masked map were calculated with `relion_image_handler`. The same soft solvent mask was also used for the calculation between EMDB-2984 and the atomic model.

## Acknowledgements

The authors would like to thank Szilárd Páll, Nikolay Markovskiy, and Mark Berger for fruitful discussions and cuda suggestions; Shintaro Aibara and Marta Carroni for providing data and early quality testing; and Toby Darling, Jake Grimmett, and Stefan Fleichmann for technical support.

## Additional information

### Competing interests
SHWS: Reviewing editor, *eLife*. The other authors declare that no competing interests exist.

### Funding

| Funder | Grant reference number | Author |
|---|---|---|
| Medical Research Council | MC UP A025 1013 | Sjors HW Scheres |
| Vetenskapsrådet | 2013-5901 | Erik Lindahl |
| Horizon 2020 | EINFRA-2015-1-675728 | Erik Lindahl |
| Swedish e-Science Research Centre | | Erik Lindahl |
| Swedish National Infrastructure for Computing | 2015/16-45 | Erik Lindahl |

The funders had no role in study design, data collection and interpretation, or the decision to submit the work for publication.

### Author contributions
DK, BOF, Conception and design, Acquisition of data, Analysis and interpretation of data, Drafting or revising the article; SHWS, EL, Conception and design, Analysis and interpretation of data, Drafting or revising the article

### Author ORCIDs
Sjors HW Scheres, http://orcid.org/0000-0002-0462-6540
Erik Lindahl, http://orcid.org/0000-0002-2734-2794

## Additional files

### Major datasets
The following dataset was generated:

| Author(s) | Year | Dataset title | Dataset URL | Database, license, and accessibility information |
|---|---|---|---|---|
| Kimanius D, Forsberg BO, Scheres SHW, Lindahl E | 2016 | RELION-2.0 reconstruction for beta-galactosidase data in EMPIAR-10061 | http://www.ebi.ac.uk/pdbe/entry/emdb/EMD-4116 | Publicly available at the EBI Protein Data Bank in Europe (accession no: EMD-4116) |

The following previously published datasets were used:

| Author(s) | Year | Dataset title | Dataset URL | Database, license, and accessibility information |
|---|---|---|---|---|
| Scheres SH | 2014 | Beta-galactosidase Falcon-II micrographs plus manually selected coordinates by Richard Henderson | https://www.ebi.ac.uk/pdbe/emdb/empiar/entry/10017/ | Publicly available at the EBI Electron Microscopy Pilot Image Archive (accession no: EMPIAR-10017) |
| Wong W, Bai XC, Brown A, Fernandez IS, Hanssen E, Condron M, Tan YH, Baum J, Scheres SHW | 2014 | Cryo-EM structure of the Plasmodium falciparum 80S ribosome bound to the anti-protozoan drug emetine | https://www.ebi.ac.uk/pdbe/emdb/empiar/entry/10028/ | Publicly available at the EBI Electron Microscopy Pilot Image Archive (accession no: EMPIAR-10028) |
| Bartesaghi A, Merk | 2015 | 2.2 A resolution cryo-EM structure | https://www.ebi.ac.uk/ | Publicly available at |

| A, Banerjee S, Matthies D, Wu X, Milne JL, Subramaniam S | of beta-galactosidase in complex with a cell-permeant inhibitor | pdbe/emdb/empiar/entry/10061/ | the EBI Electron Microscopy Pilot Image Archive (accession no: EMPIAR-10061) |
|---|---|---|---|

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

**Appendix 1**

## Difference calculation

### Exhaustive coarse sampling

To align each particle image against a reference model, RELION performs an exhaustive grid search of a large number of reference model projections against a number of image translations. In our implementation, this difference-calculation kernel is by far the most significant computational load during classification and refinement. The computations are performed in Fourier space (*Figure 1*), where projections are interpolated slices of the Fourier transform of the reference model. The tri-linear interpolation of the Fourier components performed during extraction of the Fourier-volume slice is both computationally heavy, and has significant latency associated with reading data. This is despite the use of texture objects, which use a dedicated cache and specialised storage formats. To improve memory usage, it is important to reuse the sliced data in the kernel. The maximum number of parallel tasks within this kernel is $P \times T \times C$, where $P$ is the number of orientations, $T$ the number translations, and $C$ the number of Fourier components (pixels). However, reuse of sliced data from the reference model requires at least one synchronisation and data communication within the $P$ groups of processes comparing translations with a common Fourier slice. To avoid performance penalties, we make sure all such groups are executed in the same thread-block to enable fast communication through shared memory.

The limitations in shared memory size are circumvented by splitting the reference slice into chunks of components (pixels) that are loaded separately, the management of which does however create further overhead. We fur- ther improve performance by grouping reference orientations (in groups of $P_0$ slices) to also enable reuse of translated image components. The number of completely parallel tasks is then in fact reduced, which may limit perfor- mance by potentially not saturating the hardware with a sufficient number of independent tasks. However, this reduction in turn reduces the number of reads of the Fourier transform of the particle image. This, and the reuse of translated components, ultimately provided a significant performance boost.

The described protocol was implemented by dividing the kernel into two stages, where reference data is prepared in the first stage and then reused as much as possible in the second. *Figure 9* illustrates this setup in detail. The two arbitrary parameters $N$ and $P_0$ can be tuned to balance the shared memory size with the reuse of the translation intermediates. To avoid overfit- ting these parameters to benchmarking systems, we have selected values that yield a suitable minimum required shared memory size for typical available hardware.

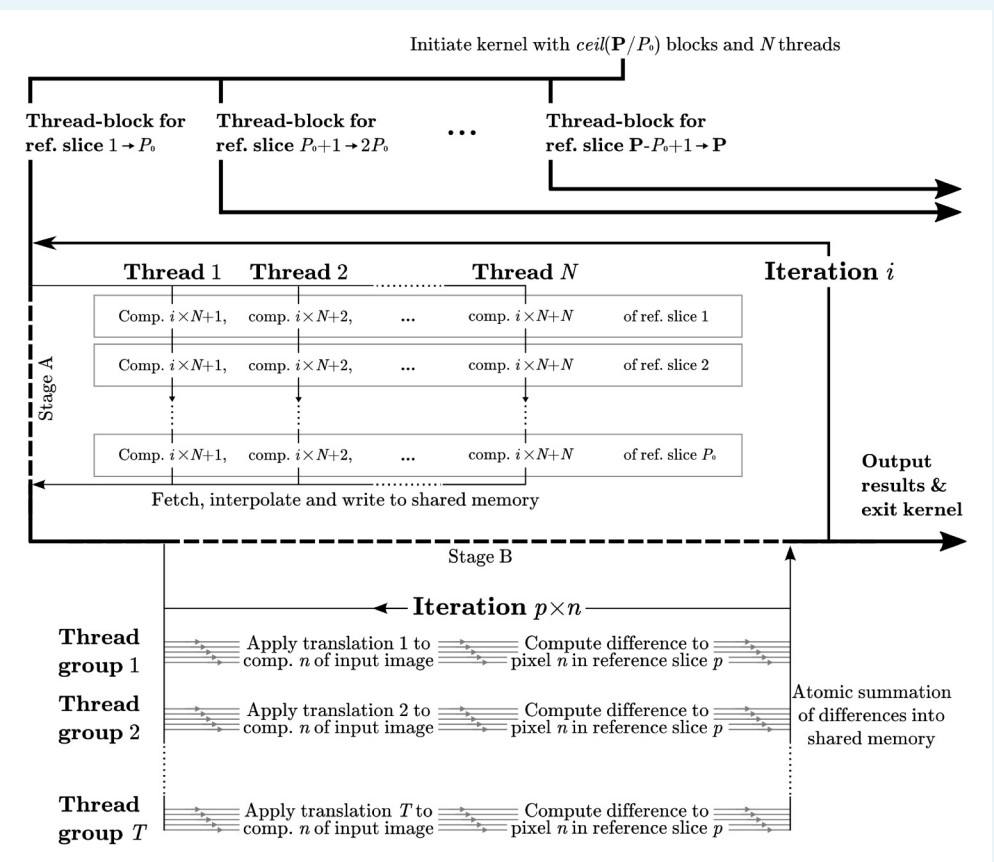

**Appendix 1—figure 1.** Computational flow in difference calculation kernel. The kernel is initiated with $ceil(\mathbf{P}/P_0)$ thread-blocks and $N$ threads, where P is the total number of projections. The work flow of a thread-block in each iteration $i$ is divided into two stages. In stage A the $N$ pixels of $P_0$ reference slices are fetched through texture memory, interpolated, and stored in shared memory. This data is then exhaustively reused in stage B, where groups of threads compute the differences to the corresponding translated image components. Individual threads within a group work with different image components, $n$, of each reference slice, $p$. Collectively all threads iterate through the $N$ components of each reference slice, for a total of $N \times P_0$ components for each iteration $i$. The final result is reduced back into shared memory through atomic reduction operations. All image components are covered as $i$ goes from 1 to $ceil(C/N)$, where C is the total number of Fourier components. A reduced sum of differences for each pair of orientation and translation is written to global memory prior to the kernel exiting.

## Sparse fine sampling

Following the exhaustive search of orientations described above, relion performs a second, fine-grained, search of the orientations which contributed most to the total alignment weight. In the majority of cases this constitutes a more sparse operation, as a few orientations and/or classifications are typically dominating. If one were to invoke the same kernel in this case, many thread-blocks would contribute insignificantly or even perform null work. We therefore chose to specialise this as a separate kernel, to reach better efficiency and stay within hardware requirements under sparse but fine sampling. The fine-grained search therefore proceeds through a preparatory stage wherein the significantly contributing combinations of reference orientations and image translations are divided into jobs (see **Figure 10**). Lists of the relevant orientations and translations are also created. A specialised kernel is then provided with this list of jobs, and invoked in parallel to execute them independently. Jobs

are also created to reuse unique information (a reference slice), and minimise overhead associated with reading relevant translation indices from high-latency global memory by using sequential translations within each job. See *Figure 10* for an in-depth description of this implementation.

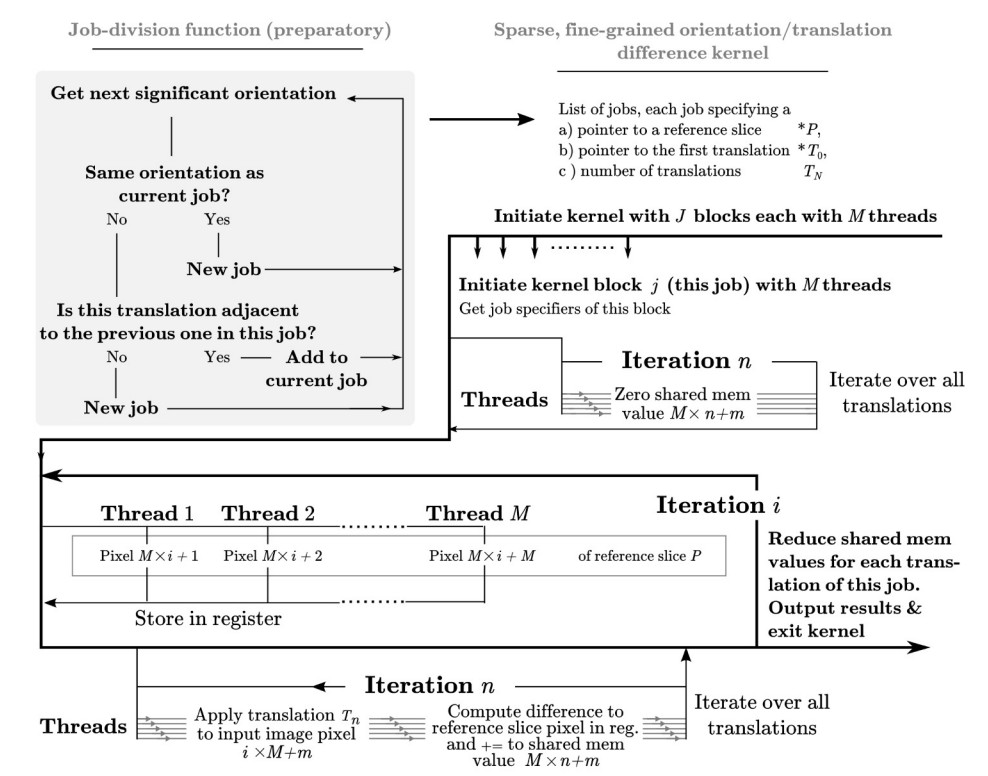

**Appendix 1—figure 2.** A dedicated kernel function performs the targeted fine-grained examination of the most significantly matching regions during image alignment against a reference model. The oversampling of each of five fitting dimensions during fine-grained search renders storage of all possible weights intractable, so input and output data are stored with explicit mapping arrays. These are read by the kernel function thread-block, rather than inferred based on block ID. This creates overhead and possible latency of global memory access, which makes this kernel even further separated from the exhaustive kernel represented in *Figure 9*. Here, a pixel-chunk of a single projection is reused for a number of sequential translations, arranged contiguously if possible. Invoking separate thread blocks for non-contiguous translations allows some implicit indexing of them, which affords better access patterns for SIMD instructions and reduced latency. Due to the sparseness, shared memory can also be used for in-kernel summation of all pixels of each image, which despite some some required explicit thread-level synchronisation increases throughput by avoiding the higher latency of atomic write operations during image summation in the coarse-search kernel.

## Appendix 2

### Back-projection

The calculated weights for different orientation of the particle image are used to back-project the image data into a 3D volume (**Figure 11A**). This subroutine takes a comparably small amount of 2D data as input and outputs it to a large container of 3D data, made up of voxels, elements much like pixels in an image. Each image pixel contributes to eight voxels through a tri-linear interpolation.

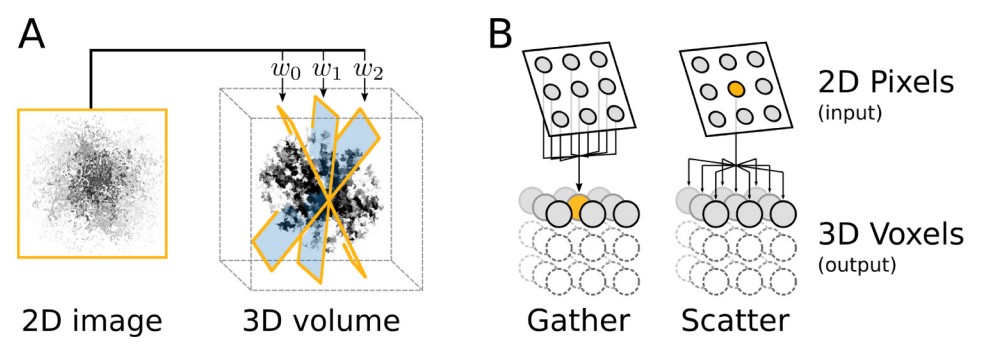

**Appendix 2—figure 1.** Computational flow of fine-grained search kernel. (**A**) Weighted back-projection of a 2D image into three different planes. We explored two memory access approaches (**B**) for this task, namely gather and scatter. In the gather approach a process (marked with orange) is assigned to individual or groups of 3D voxels. The process read from the input image and updates the data of the assigned voxel(s).

Two fundamentally different implementations were explored for this task, using either gather or scatter memory access patterns (**Figure 11B**). The issue of parallel processes writing to the same memory position simultaneously is avoided in the gather approach by restricting subsets of positions in the output to individual processes. In the scatter approach, on the contrary, write clashes are handled with atomic writes, where memory positions are reserved just in time prior to the write operation. This extra set of operations have an overhead and can become a considerable performance issue in regions with many clashes, e.g. close to the origin. However, the scatter approach enables full reuse of the interpolation of pixels for all the affected voxels, since no access restrictions exists for write operations.

In the scatter approach, processes are instead assigned to individual or groups of pixels and can output to all of the voxels. In our benchmarks the scatter approach performed significantly better and was thus selected as the standard. This is most likely due to the fact that individual images on average contribute very sparsely to the 3D volume and hence only affect a small subset of voxels. Since the number of processes in the gather approach is proportional to the number of voxels, this renders many initiated processes jobless. Both this issue and the reduced reuse of intermediates can be addressed to some degree by enabling processes to manage groups of voxels. This method yielded some performance improvements, but the scatter approach nevertheless provides superior performance.

## Appendix 3

### Hardware recommendations

Each new generation of GPU hardware has provided significant performance gains and increasing amounts of memory, so the most important factor is to use a new GPU – currently Pascal-class cards. Titan-X (Pascal) cards are likely to provide the highest performance, but due to the substantial price premium of these cards, we believe GTX 1080 cards offer better value. The workstation with quad GTX 1080 GPUs described in the materials can currently be assembled from parts for around $5000 (See links from the relion home page), or purchased pre-assembled from a few vendors for about $6200. Since the heavy computational steps run on the GPU, there is no need for dual CPU sockets. However, disk I/O bandwidth needs are high, and as seen in *Figure 4C* it helps to use two SSDs configured in RAID0. This provides a powerful cluster replacement for many users, but the quad GPU configuration is sensitive to specific motherboard models and the fans can be a bit loud for a normal office environment. In our labs, the currently most popular option is rather a dual-GPU desktop that is both quiet and smaller. This will work with virtually any motherboard with two GPU sockets. With the cheap GTX 1070 cards the machine parts only cost $2000, and we expect it can be obtained from vendors for roughly $2500. A single SSD is sufficient in this case. All these prices are subject to fluctuations (and changes due to new hardware), but it is nevertheless interesting to compare the performance/price for the most computationally intensive 3D classification with the x86 CPU cluster ($\sim$$8500/node, not including fast network). Based on the run times in *Figure 4C*, the performance/price ratio is a factor 27 higher for the quad-GPU workstation (with RAID0 SSD) compared to x86 CPU nodes, and a factor 45 higher for the low-cost dual GTX 1070 desktop (at the cost of lower absolute performance). This is based on pre-assembled system prices, and while some vendors might charge more there is always the option to assemble systems from parts for even better value. When it comes to rack-mounted alternatives, many such nodes only work with professional-class (Tesla) cards, but there are larger 4U nodes that work with quad GTX 1080 cards, and even a few 1U models. However, this is highly dependent on the hardware and the reader is advised to seek up-to-date recommendations from vendors or the RELION web pages.

