## [Decision Letter]

Thank you for submitting your article "Accelerated cryo-EM structure determination with parallelisation using GPUs in RELION-2" for consideration by *eLife*. Your article has been reviewed by four peer reviewers, one of whom, Sriram Subramaniam, is a member of our Board of Reviewing Editors, and the evaluation has been overseen by John Kuriyan as the Senior Editor. The following individuals involved in review of your submission have agreed to reveal their identity: Steven J Ludtke and J Bernard Heymann. A further reviewer remains anonymous.

Following extensive discussions with the reviewers, we are willing to re-review a revised version of your manuscript. The revised version must address all of the technical comments from the reviewers and include a substantive technical section with more details of the software implementation, as well as a more extensive description of suitable platforms with a credible comparison of performance.

The main points of this discussion are noted below:

1) There was agreement that the main point of this manuscript was to advertise the existence of a GPU port of Relion, which many people in the field will likely wish to use. The changes to Relion described in the manuscript do not improve the scientific results, but improve computational efficiency that could potentially accelerate discoveries. Nevertheless, the reviewers expressed doubts that the increase in computational efficiency would stand out as a significant advance technologically or methodologically.

2) The reviewers also noted that the extent of the performance improvement with GPU was exaggerated and potentially misleading; this is an important parameter because this would lead to users making purchasing decisions without the correct facts. While the reviewers felt it is not possible to be completely equitable when comparing completely different hardware platforms, the views was that this manuscript is more strongly skewed towards GPU for the following reasons:

a) Very different generations of hardware were compared (the CPUs are 4 years older than the GPUs, and many changes have been made to CPUs in that time)

b) The code in Relion 2.0 was substantially optimized in the GPU version; it seems that many of these optimizations could be easily included in the CPU version of Relion 2.0. A better comparison would be to compare equivalent generations of GPU/CPUs, and equivalent versions of Relion.

3) There was concern that the specific improvements to the code that helped improve GPU performance were not identified clearly and that it was difficult to tell which are only of interest for GPU and which can be implemented for CPU. A portion of the community may have strong interest in the details of these improvements, while others will likely only be interested in the question of which platform is more likely to be cost efficient and remain relevant over a relatively extended lifetime. However, this manuscript does not lay out these improvements clearly for the first group, but also does not provide fully controlled comparison of comparable platforms for the second group.

The suggestions from the reviewers follow:

1) The manuscript needs to include a paragraph discussing previous EM work that used GPU implementations, encompassing the references listed below. Pubmed /26592709 is cited in section 2.1 in another context, but not for GPU implementation.

http://www.ncbi.nlm.nih.gov/pubmed/20538058http://www.ncbi.nlm.nih.gov/pubmed/24060989http://www.ncbi.nlm.nih.gov/pubmed/26370395http://www.ncbi.nlm.nih.gov/pubmed/23644547http://www.ncbi.nlm.nih.gov/pubmed/20558298http://www.ncbi.nlm.nih.gov/pubmed/26592709

2) It would be beneficial to include significantly more algorithmic details in the paper. The way it is currently written seems to present mainly the benchmark tests, while leaving largely absent the explanation of what was important for the development of the GPU code.

3) On a technical note, regarding the experiments made for comparing CPU and GPU code and between single and double precision, one would assume that the authors used the same initial model, parameters and random seed in all cases (--random_seed option in RELION) to initialize all the runs, but this is not mentioned anywhere in the text. If this was actually done, a fairly thorough analysis of the assigned orientations and orientational probabilities could've been done where the single difference between the runs would've been the different program implementations.

4) In paragraph one of the Results section, the mention of what the authors believe to have the most prominent influence on performance is only vaguely explained:

"To overcome this, our implementation instead stores oversampled Fourier transforms of every class-reference in GPU memory, and extracts 2D-slices (in any orientation) on demand. By utilising fast-access data structures known as textures (normally used to project images on 3D objects), on-demand projection in fact achieves faster execution compared to reading pre-calculated projections from memory."

i) The authors claim to store oversampled Fourier transforms of every class-reference in GPU memory. What kind of transforms are being referred to, 3D or 2D? If this is a 3D oversampled FFT volume, then it should be mentioned and the sentence needs to be corrected to have one FFT transform per class-reference. On the other hand, if the authors mention 2D reference FFT slices (which is less likely because it is not consistent with the on-demand projection statement), they need to elaborate on how many such slices they used and on angular sampling strategies.

ii) The authors need to expand upon using texture operations for the on-demand projection. Specific citation(s) would be helpful.

5) There are inconsistent statements regarding the differences between the methods used in RELION 1.4 and the new accelerated version of RELION. On one side the authors state: "Neither method nor behaviour of RELION has changed from that of version 1.4.", but later on they describe algorithmic changes to the particle picking strategy in Section 2.2. The only tangible difference appears to be the discarding of resolution information in micrographs beyond that of search templates, but this has to be properly highlighted and the statement in Section 2.1 revised to reflect the fact that there are changes, albeit subtle, between versions 1.4 and 2.0 other than the GPU speedup. It will save end-users time and confusion if all of the changes are documented.

Is it correct that Relion adopts a maximum-likelihood approach, and not a Bayesian approach because all the priors (distributions) it uses are refined? It would be good that a person familiar with the difference between "Bayesian" and "Maximum Likelihood" could clarify this point to get it right in the published version.

6) There are several comparisons noted in the paper that need to be clarified.

i) The comparison draw between overall processing time and the time it took the authors to download the data from EMPIAR is meaningless, there are many factors that can influence data download speed of large datasets and to use that as a benchmark can be misleading. The typical RELION user will probably not be downloading data from a database for processing but instead would be collecting and processing their own data. A comparison against the time it takes to collect data would therefore be more appropriate and provide the reader with a better reference point about what the other rate limiting steps in single particle cryo-EM. Another useful comparison is to estimate the number of CPUs required to process the data in the same time (115h) that it took for a GPU-driven solution.

ii) Distributing a job across a cluster leads to substantial overhead (e.g. network traffic) that will not be present when running on a single workstation. While it may be justified to look at the overhead as one reason for the slower execution of jobs on CPU systems, this should ideally be separated in benchmark tests from the actual CPU hours needed to complete the job. Maybe some of the tests could be rerun on a single workstation to exclude cluster-related overhead.

7) It is not clear who might be the intended reader of the manuscript. It is clearly not meant for an average user of Relion package, who most likely is a structural biologist oblivious to intricacies of computer code and its dependence on hardware platform. The problems mentioned are immediately apparent at the beginning of the Results section. The fact that "class" is used to mean "3D structure" took me by surprise, particularly that one has to compare Figure 1 with Figure 2 to realize that. Some formulations are cryptic – "Even individual image pixels are evaluate independently from on another". Specifically how one evaluates individual pixels, for what purpose, and why doing it independently would constitute an advantage?

Results section, first paragraph; what is "class-reference"? What follows is difficult to understand – if, as stated, 2D slices are extracted from a presumably 3D volume, this constitutes a projection operation, but next sentence casts it in doubt, as it is stated this is being done using textures "normally used to project images on 3D objects". So it is not a projection? Or some other projection is meant?

Subsequently, much is made of the number of classes, but it is not clear what these classes are and why one would want to have them. How their number is decided? What if one wants to preform analysis without any classes, is it possible? What would be the speed up?

The most cryptic is the closing sentence of the section: Neither method nor behaviour of relion has changed from that of version 1.4. First, what is so special about version 1.4? How a casual reader is supposed to know that or worse, care about this number? It would also appear the statement is contradicted by what follows, as much is made of single precision GPU implementation as opposed to double precision of the general purpose original code. On a side, why double precision is "so-called"? I am not sure what is meant here.

Finally, section on Limited precision does not offer much outside of generalities and ends at a surprising statement: "Execution of the iterative gridding algorithm.… appeared to show significant loss of information." Appeared? How carefully was it investigated? What is "significant"?

8) Figures are microscopic, which probably reflects authors' interest in atomic-scale objects, but at the same time they are filled with details essential for the understanding of the text. In the very least, they should be properly referenced in the text in key places. The final maps should also be deposited as is customary in the field, this intent is not indicated in the manuscript.

9) As this manuscript may entice people to spend tens to hundreds of thousands of dollars in research funds, it is incumbent upon the authors and reviewers to insure that this study is as balanced and fair as possible. There was a similar pro-GPU move ~8 years ago, Frealign, EMAN2 and SPARX were all parallelized for the GPU, achieving typically 20-80x speedups, and many purchased hardware, but few made effective use of it for very long. While the reviewers do not dispute the accuracy of the presented results, they believe that the results are highly biased based on the specific hardware selection, and on issues not discussed in the manuscript.

10) There is some price/performance advantage in GPUs, but the current estimate is simply ludicrous, and many people reading this manuscript will not understand the technical differences. Primary issues that occur immediately are:

– The processors used on the 120 core test is a 4 year-old processor considered "end of life" by Intel. These are being compared to a set of 4 GPUs so new they are difficult to acquire at most vendors. The vectorization and architecture changes made by intel in recent years, major hyperthreading improvements, and other factors will come into play. Specifically:

–- CPU optimization for the specific processors in use? Image processing code compiled for specific current generation CPUs with -march=native often see 30% or more speedups, without any coding changes.

–hyperthreading use. This likely will not have an impact the older processors in the 120 core test, but Xeon v3 processors can often see speedups of ~30% even on compute-intensive loads with moderate thread oversubscription.

– Were the core parallelism improvements made for the GPU (those which were applicable) also compiled into the CPU version?

– Running on 120 cores implies use of MPI, and brings many other factors (network, available I/O) into play when comparing speed.

– Was the CPU computation done with single precision at similar points to make the comparison fair?

In Figure 4, the authors are claiming a 10-40x price-performance improvement. One reviewer notes that he/she configured two machines at a reputable, but inexpensive vendor. Both machines had 64 GB of RAM, a 1.2 TB SSD. One was configured with 4x 1080 GPU cards and 8 CPU cores at 2.6 GHz. The other was configured with a base-level GPU for display and 2x E5-2697Av4 -> 32 cores @ 2.6 GHz. Both machines come in at ~$9000. As compared to their "one node", this machine has 32 rather than 12 cores, and with improvements in the CPU, each is ~1.5x more powerful than their test machine. This would already bring their price/performance comparison down by 4x, ignoring other issues.

11) The reviewers do not believe the current comparison in the manuscript is fair without at least some attempt to test on current generation CPUs with some real attempt to insure things are compiled and executed optimally. This need not be a month-long test, but at least a few iterations of a single model refinement to establish a baseline for modern hardware. As the major point of the manuscript is benchmarking, this is a minimum requirement for publication.

12) The following statements are not clear:

"In fact, using GPU-enabled relion, the time needed for classification is only weakly dependent on the number of classes used, whereas the CPU-based implementation has a much steeper linear dependence (Figure 3)"

and:

"In the CPU-only version, the computational time scales linearly with increased number of classes (Figure 3) due to the serialised single image calculations, whereas GPU-enabled execution can show better-than linear scaling."

Looking at Figure 3, this is not clear. The GPU curve also visually appears to be following a linear trend, and rough estimates show a similar scaling to the GPU curve.

The values on the GPU curve are small enough compared to the size of the dots, this is difficult to assess quantitatively, but this figure certainly doesn't illustrate the point being made.

13) In Figure 3, the prominent lump in the middle is quite odd. The statement "GPU's ability to parallelise the drastically increased number of tasks" does not really explain it. If N independent tasks need to be completed in iteration 10, and 30N need to be completed in iteration 15, the only way this curve makes sense is if the GPU is massively underutilizing resources during iteration 10. It also would make no sense that going from 7.5 to 3.8 degree sampling would increase the number of tasks 30-fold. Something very peculiar is going on here, and it seems important to understand it.

"While relion-1.4 only exploited parallelism over images (left), in the new implementation classes and all orientations of each class are expressed as tasks that can be scheduled independently on the accelerator hardware (e.g. GPUs)"

Could the same change not be tested on CPUs as well?

14) Figure 4 – Why is this test suddenly shown for a ribosome? How does this compare on the benchmark ΒGal data set?

15) The use of GPU's is usually problematic because of rapid progress in GPU technology. This means that many different GPU's are available and in particular older one's suffer from poor feature support. The authors need to explain exactly what is required in a GPU and which GPU's are too old or inadequate.

16) The loss of information in the gridding reconstruction algorithm using single precision is not explained clearly. Typically, this indicates very small or very large values that could be avoided by some normalization approach. This could also indicate that the iterative refinement in the gridding algorithm is too fast, generating instabilities with single precision.

17) In the Introduction, the authors mention some other software packages giving the impression of a comprehensive list. However, some software packages are left out. Either be comprehensive or be more general in referring to the wiki showing software packages.

18) "This allows high-resolution structure determination with minimal bias or user input." The impression is that a Bayesian approach is less biased than other methods. This is incorrect, as Bayesian methods are just as biased as any other when it is fed erroneous data (such as poorly picked particles) or an inappropriate starting reference (prior) is used.

19) There is no information about the availability of the package (web site and license) or about deposition of the maps.

---

## [Author Response]

*The main points of this discussion are noted below:*

*1) There was agreement that the main point of this manuscript was to advertise the existence of a GPU port of Relion, which many people in the field will likely wish to use. The changes to Relion described in the manuscript do not improve the scientific results, but improve computational efficiency that could potentially accelerate discoveries. Nevertheless, the reviewers expressed doubts that the increase in computational efficiency would stand out as a significant advance technologically or methodologically.*

As previously discussed, the acceleration described in our paper has the potential to completely change how we work with image reconstruction. With the cost of image processing substantially reduced, more extensive classification strategies, for example, may now be explored. Because this now becomes possible even for labs that do not have access to large supercomputing facilities, our approach will also contribute to increasing the accessibility of high-resolution cryo-EM structure determination to new users. In the revised version, we now also describe the modifications to our algorithms on a level where others can build on them. We have extended the manuscript with a more detailed methods section, descriptions of how the various steps have been tailored to modern GPU hardware, and we also provide appendices with detailed descriptions of the technical implementations, as requested by the reviewers.

*2) The reviewers also noted that the extent of the performance improvement with GPU was exaggerated and potentially misleading; this is an important parameter because this would lead to users making purchasing decisions without the correct facts. While the reviewers felt it is not possible to be completely equitable when comparing completely different hardware platforms, the views was that this manuscript is more strongly skewed towards GPU for the following reasons:a) Very different generations of hardware were compared (the CPUs are 4 years older than the GPUs, and many changes have been made to CPUs in that time)b) The code in Relion 2.0 was substantially optimized in the GPU version; it seems that many of these optimizations could be easily included in the CPU version of Relion 2.0. A better comparison would be to compare equivalent generations of GPU/CPUs, and equivalent versions of Relion.*

Yes, we previously discussed this comparison should have used the latest-generation hardware. We have rewritten the text and base all comparisons on brand-new Xeon E5- 2960v4 nodes. However, because of the higher cost for such nodes, because more cores share memory, and because we introduced further optimizations in the GPU code, the relative performance advantages remain. When comparing to even more cost-efficient GTX 1070 GPUs that we now include in the performance/price ratio can even get close to a factor 50. However, we are not interested in arguing specific numbers (that no doubt will change), so to make the manuscript relevant in the long run we have also altered the formulations in the Abstract and understate it to only mention order-of-magnitude advantages.

We have also included detailed specifications of hardware, and provide users with examples of both cost-efficient and performance-leading hardware (in appendix III).

We are equally interested in improving CPU performance, and will invest efforts there in the future, but as seen e.g. from efforts by Intel to accelerate RELION on Xeon Phi (that only provide 30% speedup), this is not trivial work.

*3) There was concern that the specific improvements to the code that helped improve GPU performance were not identified clearly and that it was difficult to tell which are only of interest for GPU and which can be implemented for CPU. A portion of the community may have strong interest in the details of these improvements, while others will likely only be interested in the question of which platform is more likely to be cost efficient and remain relevant over a relatively extended lifetime. However, this manuscript does not lay out these improvements clearly for the first group, but also does not provide fully controlled comparison of comparable platforms for the second group.*

We were initially hesitant about making the manuscript too technical in the initial version, but have now rewritten the methods section to provide a better technical background to the algorithms used and where the bottlenecks are, how this maps to GPU hardware features, and we added two appendices with specific details of the kernel implementations for the technically interested readers. We think that this has improved the paper substantially for both audiences, and thank the reviewers for their suggestion.

The suggestions from the reviewers follow:

*1) The manuscript needs to include a paragraph discussing previous EM work that used GPU implementations, encompassing the references listed below. Pubmed /26592709 is cited in section 2.1 in another context, but not for GPU implementation.*

http://www.ncbi.nlm.nih.gov/pubmed/20538058http://www.ncbi.nlm.nih.gov/pubmed/24060989http://www.ncbi.nlm.nih.gov/pubmed/26370395http://www.ncbi.nlm.nih.gov/pubmed/23644547http://www.ncbi.nlm.nih.gov/pubmed/20558298http://www.ncbi.nlm.nih.gov/pubmed/26592709

We’re delighted to add all of them, and these contributions are now mentioned in the Introduction.

*2) It would be beneficial to include significantly more algorithmic details in the paper. The way it is currently written seems to present mainly the benchmark tests, while leaving largely absent the explanation of what was important for the development of the GPU code.*

In the revised manuscript we have added a Method section which more thoroughly introduces parallelism and which goes on to explain the fundamental aspects of the acceleration. Additionally, appendices are provided and referred to in the text, which explain and schematically show program flow and discusses effects on performance.

*3) On a technical note, regarding the experiments made for comparing CPU and GPU code and between single and double precision, one would assume that the authors used the same initial model, parameters and random seed in all cases (--random_seed option in RELION) to initialize all the runs, but this is not mentioned anywhere in the text. If this was actually done, a fairly thorough analysis of the assigned orientations and orientational probabilities could've been done where the single difference between the runs would've been the different program implementations.*

All experiments were performed using the same command-line, however the non- deterministic nature of the GPU-enabled execution provides an element of variability which to some extent makes identical seeds superfluous. We nonetheless agree that this is an important consideration to mention, and this is now explicitly done in the revised manuscript. The analysis of the assigned orientations is a good suggestion. We've added histograms of the differences in orientations as Figure 5—figure supplement 1.

*4) In paragraph one of the Results section, the mention of what the authors believe to have the most prominent influence on performance is only vaguely explained:*

*"To overcome this, our implementation instead stores oversampled Fourier transforms of every class-reference in GPU memory, and extracts 2D-slices (in any orientation) on demand. By utilising fast-access data structures known as textures (normally used to project images on 3D objects), on-demand projection in fact achieves faster execution compared to reading pre-calculated projections from memory."*

*i) The authors claim to store oversampled Fourier transforms of every class-reference in GPU memory. What kind of transforms are being referred to, 3D or 2D? If this is a 3D oversampled FFT volume, then it should be mentioned and the sentence needs to be corrected to have one FFT transform per class-reference. On the other hand, if the authors mention 2D reference FFT slices (which is less likely because it is not consistent with the on-demand projection statement), they need to elaborate on how many such slices they used and on angular sampling strategies.*

For 3D refinements or 3D classification, we keep oversampled 3D Fourier transforms in memory and take 2D slices out of these on demand. This is covered more in-depth by the expanded method-section. The phrase “Instead, like the CPU code, our implementation stores a two-fold oversampled Fourier transform of each reference in GPU memory, and 2D slices (along any orientation) are extracted only when needed.” has also been adopted to clarify storage of transformed volumes.

*ii) The authors need to expand upon using texture operations for the on-demand projection. Specific citation(s) would be helpful.*

We now both explain textures, texture units, and how it can be used to improve performance, in the expanded methods section and appendix I.

*5) There are inconsistent statements regarding the differences between the methods used in RELION 1.4 and the new accelerated version of RELION. On one side the authors state: "Neither method nor behaviour of RELION has changed from that of version 1.4.", but later on they describe algorithmic changes to the particle picking strategy in Section 2.2. The only tangible difference appears to be the discarding of resolution information in micrographs beyond that of search templates, but this has to be properly highlighted and the statement in Section 2.1 revised to reflect the fact that there are changes, albeit subtle, between versions 1.4 and 2.0 other than the GPU speedup. It will save end-users time and confusion if all of the changes are documented.*

The conflicting statements have been resolved by removing the former. All implemented changes have been described and documented.

*Is it correct that Relion adopts a maximum-likelihood approach, and not a Bayesian approach because all the priors (distributions) it uses are refined? It would be good that a person familiar with the difference between "Bayesian" and "Maximum Likelihood" could clarify this point to get it right in the published version.*

This is not entirely correct, although much of this discussion is semantics. RELION uses a regularised likelihood approach, where a likelihood function is complemented with a prior probability of the model. According to Bayes' law, the regularised likelihood function is proportional to the posterior probability of the model given the experimental data. Because RELION estimates the parameters of the prior from the data themselves, this type of approach is called an empirical Bayesian approach (see Wikipedia for more details: https://en.wikipedia.org/wiki/Empirical_Bayes_method). Note that un-regularised likelihood optimisation algorithms (like e.g. implemented in Xmipp, and in a limited manner also in Frealign) do not include a prior on the model. These approaches are also called "maximum likelihood", but often depend on more ad-hoc regularisation approaches, for example through customised low-pass filters. To distinguish RELION from the unregularised likelihood approaches, we use the term "Bayesian". In response to the reviewer's comment, we've added the distinction of "empirical" Bayesian to the revised manuscript, and explain the entire algorithm better in the methods section.

*6) There are several comparisons noted in the paper that need to be clarified.*

*i) The comparison draw between overall processing time and the time it took the authors to download the data from EMPIAR is meaningless, there are many factors that can influence data download speed of large datasets and to use that as a benchmark can be misleading. The typical RELION user will probably not be downloading data from a database for processing but instead would be collecting and processing their own data. A comparison against the time it takes to collect data would therefore be more appropriate and provide the reader with a better reference point about what the other rate limiting steps in single particle cryo-EM. Another useful comparison is to estimate the number of CPUs required to process the data in the same time (115h) that it took for a GPU-driven solution.*

This is a valid objection. This data set comprises more than 1,500 micrographs, and although no exact timings for the data acquisition are given in the corresponding paper, it is likely that the data acquisition took multiple days. Therefore, we've replaced this statement and instead state that the total processing time was probably comparable to the data acquisition time.

Regarding the second point and quantitative comparisons of CPU and GPU hardware: such comparisons are given in detail for the ribosome benchmarks, and Figure 4 as well as appendix III now provide both performance and performance/price ratios for different GPU options. The section on the betagal data set is rather intended as a representative use case where we illustrate that it is now indeed possible to process even one of today’s largest data set on a single GPU workstation.

*ii) Distributing a job across a cluster leads to substantial overhead (e.g. network traffic) that will not be present when running on a single workstation. While it may be justified to look at the overhead as one reason for the slower execution of jobs on CPU systems, this should ideally be separated in benchmark tests from the actual CPU hours needed to complete the job. Maybe some of the tests could be rerun on a single workstation to exclude cluster-related overhead.*

Because the absolute speed of the CPU implementation is so much lower, presently the network overhead is not a major limiting factor; when comparing performance for 1 vs. 10 CPU nodes in our benchmark, our communication overhead during the expectation step was below 5% when using 10Gb network. A comparison of CPU and GPU implementations on a single desktop would only serve the purpose of illustrating that the CPU implementation is entirely unfeasible to run on desktops (it would take over a month), whereas it works great with GPUs. Therefore, a limited amount of network overhead is inescapable for realistic processing using CPUs, whereas it can be avoided using GPUs.

However, doing extensive MPI benchmarking fairly would require an entire paper of its own, so we have instead removed our statement about MPI overhead.

*7) It is not clear who might be the intended reader of the manuscript. It is clearly not meant for an average user of Relion package, who most likely is a structural biologist oblivious to intricacies of computer code and its dependence on hardware platform. The problems mentioned are immediately apparent at the beginning of the Results section. The fact that "class" is used to mean "3D structure" took me by surprise, particularly that one has to compare Figure 1 with Figure 2 to realize that. Some formulations are cryptic – "Even individual image pixels are evaluate independently from on another". Specifically how one evaluates individual pixels, for what purpose, and why doing it independently would constitute an advantage?*

We now include more algorithmic detail in the expanded methods section, and we also add more useful information on the different hardware platforms for the average RELION user (Figure 4, appendix III). We agree that the modern structural biologist interested in doing cryo-EM does need some minimum knowledge about issues pertaining to high- performance computing in order to make the best use of the available hardware (using RELION or any other program). By adding more information about the different aspects of the computations in our implementation, the revised manuscript should also be useful in that respect. The formulation of pixel “evaluation” has also been revised to more directly indicate any logical arithmethical operation.

*Results section, first paragraph; what is "class-reference"? What follows is difficult to understand – if, as stated, 2D slices are extracted from a presumably 3D volume, this constitutes a projection operation, but next sentence casts it in doubt, as it is stated this is being done using textures "normally used to project images on 3D objects". So it is not a projection? Or some other projection is meant?*

*Subsequently, much is made of the number of classes, but it is not clear what these classes are and why one would want to have them. How their number is decided? What if one wants to preform analysis without any classes, is it possible? What would be the speed up?*

We have removed the term class-reference from the manuscript. We now speak about "3D reference maps, or 3D classes" (e.g. subsection “Regularised likelihood optimisation”) to make this clearer. The number of classes to refine is a user-defined parameter, as is the case with most multi- reference refinement programs in the field.

*The most cryptic is the closing sentence of the section: Neither method nor behaviour of relion has changed from that of version 1.4. First, what is so special about version 1.4? How a casual reader is supposed to know that or worse, care about this number? It would also appear the statement is contradicted by what follows, as much is made of single precision GPU implementation as opposed to double precision of the general purpose original code. On a side, why double precision is "so-called"? I am not sure what is meant here.*

*Finally, section on Limited precision does not offer much outside of generalities and ends at a surprising statement: "Execution of the iterative gridding algorithm.… appeared to show significant loss of information." Appeared? How carefully was it investigated? What is "significant"?*

Version 1.4 is simply the most recent stable version of RELION. We've clarified this in the revised manuscript by instead referring to 'previous versions' of RELION. We have also removed the “so-called” statement regarding double precision. With regards to the gridding algorithm, we felt it necessary to briefly comment on what was omitted from GPU acceleration, but felt it unnecessary to elaborate on changes not made. In the revised manuscript we have removed the "significant" statement, and now mention the gridding algorithm as an opportunity for further improvement.

*8) Figures are microscopic, which probably reflects authors' interest in atomic-scale objects, but at the same time they are filled with details essential for the understanding of the text. In the very least, they should be properly referenced in the text in key places. The final maps should also be deposited as is customary in the field, this intent is not indicated in the manuscript.*

We don't really understand how this happened (on our screen/prints the figures aren't microscopic at all). Regardless, we apologize for any inconvenience caused. All figures are vector based and will be rescaled to balance layout and readability, and we have increased the relative size of small type and objects. We have deposited the β- galactosidase map at the EMDB (EMD-4116). We chose not to deposit the ribosome reconstruction from Figure 5 (as numbered in the revised manuscript), as it is basically identical to the EMDB entry 2660, so would only constitute a duplication of an existing entry.

*9) As this manuscript may entice people to spend tens to hundreds of thousands of dollars in research funds, it is incumbent upon the authors and reviewers to insure that this study is as balanced and fair as possible. There was a similar pro-GPU move ~8 years ago, Frealign, EMAN2 and SPARX were all parallelized for the GPU, achieving typically 20-80x speedups, and many purchased hardware, but few made effective use of it for very long. While the reviewers do not dispute the accuracy of the presented results, they believe that the results are highly biased based on the specific hardware selection, and on issues not discussed in the manuscript.*

There are two aspects to this point. First, there is the point of hardware selection. Second, the point of long-term usefulness of the code.

To address the first point, we have expanded the comparison of different combinations of hardware as a part of Figure 4 (and appendix III). This comparison is not a comprehensive study of all hardware available on the market, but rather illustrates the considerations relevant for modern GPUs. We believe that the relative wide spread of difference GPUs (and CPUs) gives potential users a reasonable insight into what hardware to buy.

Regarding the second point, the impact of any software implementation depends on long- term maintenance, and that the code is incorporated in the main codebase rather than a separate spin-off from an old release. This is one key reason why everyone uses GPU for some cryo-EM programs like Gctf and Motioncorr (1 or -2), but not e.g. Frealign. We note here that our implementation represents a close collaboration between the Scheres and Lindahl groups, and is not a one-off optimization of a specific application version, but a long-term commitment to GPUs for RELION, now a part of the main codebase. We are ourselves also making large hardware investments in GPUs. In addition, after having supported GPUs for close to a decade in Gromacs, the molecular simulation users have switched entirely to GPUs. We believe this backs up our claims to long-term commitment.

*10) There is some price/performance advantage in GPUs, but the current estimate is simply ludicrous, and many people reading this manuscript will not understand the technical differences. Primary issues that occur immediately are:*

*– The processors used on the 120 core test is a 4 year-old processor considered "end of life" by Intel. These are being compared to a set of 4 GPUs so new they are difficult to acquire at most vendors.*

We understand these concerns and have updated benchmarks to use newer CPU hardware. However, not all benchmarks have seen an improvement in favour of the CPU, when considering price/performance, and the cheap GTX 1070 cards provide even better performance/price ratios. Still, despite this we have decided to tone down the statements and no longer mention specific numbers in the Abstract to focus on principles, not numbers.

*The vectorization and architecture changes made by intel in recent years, major hyperthreading improvements, and other factors will come into play. Specifically:*

*– CPU optimization for the specific processors in use? Image processing code compiled for specific current generation CPUs with -march=native often see 30% or more speedups, without any coding changes.*

We used hyper-threading and the fastest (as far as we know) flags for the x86 code (RELION uses –O3 by default, which includes –march=native). Intel has also worked to accelerate RELION-1.4 for their latest Xeon Phi processors. When changing the code and selecting the best compiler flags, they see a 20-30% performance improvement – but only when adding a $4800 accelerator (the cost per node would be much higher than our nodes above). While this indicates possible improvements on the CPU-side, they do not amount to that estimated by the reviewer.

*– hyperthreading use. This likely will not have an impact the older processors in the 120 core test, but Xeon v3 processors can often see speedups of ~30% even on compute-intensive loads with moderate thread oversubscription.*

As mentioned above, hyperthreading was of course utilized for all CPU runs as is now clarified by “While we refer to the physical core count when describing hardware, hyperthreading was enabled and used for all benchmarks since it improves performance on the CPU side slightly.”

*– Were the core parallelism improvements made for the GPU (those which were applicable) also compiled into the CPU version?*

Some concepts – such as partial use of single precision as in the GPU implementation – were tested on the CPU but made execution slower (extra conversions), and thus they were not applied in the code nor in any benchmarks. Others – e.g. the autopick filtering – were clearly benefitial and were thus implemented on both the GPU and CPU code-bases. More generally, the extraction of large amounts of parallelism is key for GPUs require some 10,000 or more independent tasks to achieve good performance, meaning that the present developments are largely unapplicable to CPUs, which rather achieve good performance with a handful of general-purpose cores. Applying the GPU-intended methods and algorithms direclty to CPUs would just cause overhead. Generalized vector-operations however currently noted as a future improvement.

*– Running on 120 cores implies use of MPI, and brings many other factors (network, available I/O) into play when comparing speed.*

The mentioned overhead is in part addressed in the updated benchmarks, where all nodes make a copy of the data to a local disk prior to the computations, thereby removing the major part of network/MPI communication cost. For our benchmark using 10 vs. 1 CPU nodes, during the expectation step we never observed communication overhead in excess of 5% (on 10GbE). However, mentioned above, there is also not much alternative to MPI when using CPU clusters since few users (in our opinion) would be willing to wait a month for the results just to achieve slightly higher efficiency by limiting it to a single node.

*– Was the CPU computation done with single precision at similar points to make the comparison fair?*

Unfortunately, the performance of the CPU version does not improve with simple translation of instructions to use single precision (mainly due to conversions to double precision in parts which require more accurate computations). We would additionally like to note that it is not merely a matter of “changing double to single”, but to make it possible to get accurate results from single precision on CPUs we would need to implement similar amendments to core algorithms as in the GPU code, and since there is little or no performance improvement that is presently of limited value. Even if this is done, the relative advantage of single is much higher on GPUs, both because of a factor ~8 higher performance and use of texture units. We have updated the text to explain this better.

*In Figure 4, the authors are claiming a 10-40x price-performance improvement. One reviewer notes that he/she configured two machines at a reputable, but inexpensive vendor. Both machines had 64 GB of RAM, a 1.2 TB SSD. One was configured with 4x 1080 GPU cards and 8 CPU cores at 2.6 GHz. The other was configured with a base-level GPU for display and 2x E5-2697Av4 -> 32 cores @ 2.6 GHz. Both machines come in at ~$9000. As compared to their "one node", this machine has 32 rather than 12 cores, and with improvements in the CPU, each is ~1.5x more powerful than their test machine. This would already bring their price/performance comparison down by 4x, ignoring other issues.*

Yes, as we acknowledge above it is more fair to use the very latest Xeon cores for a comparison, and we now have access to such nodes. However, as seen from Figure 4, with the latest code version running on brand-new Xeon cores vs. GTX1080 the difference persists – and then we have not even included the 25% faster Titan-X Pascal GPUs.

Rather than discussing how theoretically “powerful” different cores might be (the reviewer e.g. does not take memory bandwidth into account), we focus on how fast the reconstruction actually runs on the hardware, taking a pragmatic view which we believe benefits users better.

There are definitely highly priced GPU systems around too, in particular when selecting dual CPU socket systems (which are not needed for RELION). We have provided publicly available (www.cryoem.se/RELION-gpu) examples of specific hardware which we have purchased, tested and re-evaluated recently, at significantly better prices. However, ultimately we do not think specific prices should be featured in a scientific paper, since they change rapidly. We think it is better to show the performance for different hardware, and encourage educated users to make their own decision based on current prices. Because we do however realize the need for some guidance on purchase options for our users, we will maintain the blog mentioned above, and have also added benchmark results with different types of hardware on the Relion WIKI page (http://www2.mrc- lmb.cam.ac.uk/relion/index.php/Benchmarks_%26_computer_hardware).

*11) The reviewers do not believe the current comparison in the manuscript is fair without at least some attempt to test on current generation CPUs with some real attempt to insure things are compiled and executed optimally. This need not be a month-long test, but at least a few iterations of a single model refinement to establish a baseline for modern hardware. As the major point of the manuscript is benchmarking, this is a minimum requirement for publication.*

Agreed – as described above, we have now completely replaced the previous benchmarks using state-of-the-art CPU hardware, which we believe backs up our statements better.

*12) The following statements are not clear:*

*"In fact, using GPU-enabled relion, the time needed for classification is only weakly dependent on the number of classes used, whereas the CPU-based implementation has a much steeper linear dependence (Figure 3)"*

*and:*

*"In the CPU-only version, the computational time scales linearly with increased number of classes (Figure 3) due to the serialised single image calculations, whereas GPU-enabled execution can show better-than linear scaling."*

*Looking at Figure 3, this is not clear. The GPU curve also visually appears to be following a linear trend, and rough estimates show a similar scaling to the GPU curve.*

*The values on the GPU curve are small enough compared to the size of the dots, this is difficult to assess quantitatively, but this figure certainly doesn't illustrate the point being made.*

The indicated statements have been reformulated to “For sufficiently large computational problems, RELION’s processing time scales linearly with increased number of classes, but since the extra calculations are much faster with the GPU- enabled version the relative advantage is larger the more classes are used (Figure 4).” This also improves the clarity of the message conveyed by the indicated figure.

*13) In Figure 3, the prominent lump in the middle is quite odd. The statement "GPU's ability to parallelise the drastically increased number of tasks" does not really explain it. If N independent tasks need to be completed in iteration 10, and 30N need to be completed in iteration 15, the only way this curve makes sense is if the GPU is massively underutilizing resources during iteration 10. It also would make no sense that going from 7.5 to 3.8 degree sampling would increase the number of tasks 30-fold. Something very peculiar is going on here, and it seems important to understand it.*

*"While relion-1.4 only exploited parallelism over images (left), in the new implementation classes and all orientations of each class are expressed as tasks that can be scheduled independently on the accelerator hardware (e.g. GPUs)"*

*Could the same change not be tested on CPUs as well?*

The reason for the “lump” is simply that the fine exhaustive search requires a huge amount of both floating-point operations and memory bandwidth, which becomes very costly on the CPU. The reason this is not evident on the GPU is that the reviewer is quite correct – we are still nowhere near utilizing the full GPU resources in the early iterations (the GPU time/iteration does go up by ~50% in this region, but the absolute number is still much lower than the CPU).

The factor 30 stems from the two-fold increase in sampling in each of the 5 fitting dimensions (3 angular and 2 translational), where the load subsequently increases by 2^5=32. Similar parallel procedures could be adopted for the CPU implementation, however the anticipated improvement does not motivate the time investment, as explained previously.

*14) Figure 4 – Why is this test suddenly shown for a ribosome? How does this compare on the benchmark ΒGal data set?*

Throughout the manuscript, we use the *Plasmodium falciparum* ribosome data set to benchmark each of the different algorithms separately (e.g. auto-picking, 3D classification and 3D auto-refinement). We have chosen this data set (for which so-called 'polished particles' from RELION's movie-processing procedure are available for download by anyone interested from the EMPIAR data base) as the standard benchmark for RELION. The betagal data set is used as an additional illustration of the impact of the GPU- implementation on an entire processing workflow. We believe that a second example data set, being the highest-resolution data set available in the EMPIAR data base, adds value to the paper. We've added the following sentence to the revised manuscript to clarify this: "This represents the largest presently available dataset in the EMPIAR database, and provides a realistic challenge."

*15) The use of GPU's is usually problematic because of rapid progress in GPU technology. This means that many different GPU's are available and in particular older one's suffer from poor feature support. The authors need to explain exactly what is required in a GPU and which GPU's are too old or inadequate.*

We understand reviewer’s concerns that rapid development of hardware can be problematic for directed optimizations of code which relies on hardware specifications. We would argue that the rapid progress in GPU technology is good and provides opportunity for long- term commitments which are able to exploit such hardware competently and without excessive optimization, of which we believe RELION is a perfect example. We do agree that information about performance and support for older cards is useful, and we have added this to the dataset and hardware section. In short, we support at least the three latest generations of NVIDIA GPUs and will continue to provide updated support and development, making the rapid progress of GPU hardware an asset rather than problematic.

*16) The loss of information in the gridding reconstruction algorithm using single precision is not explained clearly. Typically, this indicates very small or very large values that could be avoided by some normalization approach. This could also indicate that the iterative refinement in the gridding algorithm is too fast, generating instabilities with single precision.*

Yes, this is an excellent point, and we already use the normalization approach in other parts of the code. Because the current implementation avoids this issue by leaving the gridding algorithm non-accelerated (i.e. run always on the CPU), we now rather mention this opportunity for future improvement in the revised manuscript, as mentioned above.

*17) In the Introduction, the authors mention some other software packages giving the impression of a comprehensive list. However, some software packages are left out. Either be comprehensive or be more general in referring to the wiki showing software packages.*

We feel it would be inappropriate not to cite at least some of the available alternative programs, so we have extended this list with several additional software packages to be more comprehensive.

*18) "This allows high-resolution structure determination with minimal bias or user input." The impression is that a Bayesian approach is less biased than other methods. This is incorrect, as Bayesian methods are just as biased as any other when it is fed erroneous data (such as poorly picked particles) or an inappropriate starting reference (prior) is used.*

Agreed. We have removed the word 'bias' from the sentence.

19) There is no information about the availability of the package (web site and license) or about deposition of the maps.

We added the following sentence to the manuscript. "RELION-2 is both open source and free software, distributed under the GPLv2 licence. It is publicly available for download through http://www2.mrc-lmb.cam.ac.uk/relion." We also provide accession code of the deposited map, and the previously available ribosome map.